# Reliable Lifelong Multimodal Editing: Conflict-Aware Retrieval Meets Multi-Level Guidance

**Qiang Zhang**[1*], **Fanrui Zhang**[1,2*], **Jiawei Liu**[1†], **Ming Hu**[3], **Junjun He**[3],
**Zheng-Jun Zha**[1]

[1]MoE Key Laboratory of Brain-inspired Intelligent Perception and Cognition, USTC
[2]Shanghai Innovation Institute
[3]Shanghai Artificial Intelligence Laboratory
{zq_126, zfr888}@mail.ustc.edu.cn  {jwliu6, zhazj}@ustc.edu.cn
ming.hu@monash.edu  hejunjun@sjtu.edu.cn

## Abstract

The dynamic nature of real-world information demands efficient knowledge editing in multimodal large language models (MLLMs) to ensure continuous knowledge updates. However, existing methods often struggle with precise matching in large-scale knowledge retrieval and lack multi-level guidance for coordinated editing, leading to less reliable outcomes. To tackle these challenges, we propose CARML, a novel retrieval-augmented editing framework that integrates conflict-aware dynamic retrieval with multi-level implicit and explicit guidance for reliable lifelong multimodal editing. Specifically, CARML introduces intra-modal uncertainty and inter-modal conflict quantification to dynamically integrate multi-channel retrieval results, so as to pinpoint the most relevant knowledge to the incoming edit samples. Afterwards, an edit scope classifier discerns whether the edit sample semantically aligns with the edit scope of the retrieved knowledge. If deemed in-scope, CARML refines the retrieved knowledge into information-rich continuous prompt prefixes, serving as the implicit knowledge guide. These prefixes not only include static knowledge prompt that capture key textual semantics but also incorporate token-level, context-aware dynamic prompt to explore fine-grained cross-modal associations between the edit sample and retrieved knowledge. To further enhance reliability, CARML incorporates a "hard correction" mechanism, leveraging explicit label knowledge to adjust the model's output logits. Extensive experiments across multiple MLLMs and datasets indicate the superior performance of CARML in lifelong multimodal editing scenarios.

## 1  Introduction

The rapid evolution of multimodal large language models (MLLMs), distinguished by the deep integration of visual and textual modalities, has significantly expanded the application boundaries of language models in real-world scenarios [1, 41, 43]. However, the static nature of pre-trained MLLMs' embedded knowledge presents a critical bottleneck in continuously evolving information environments [7]. To address this, knowledge editing [37, 38] has emerged as a promising paradigm for efficiently updating or correcting specific knowledge within models, avoiding the high costs of retraining. This technique plays an indispensable role in critical fields such as privacy preservation [19], bias mitigation [22], jailbreak attack defense [16], and hallucination correction [44, 8].

---

*Equal contribution. † Corresponding author.

39th Conference on Neural Information Processing Systems (NeurIPS 2025).

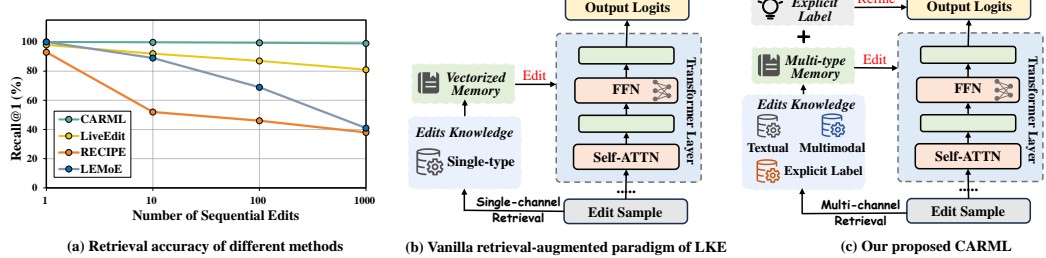

(a) Retrieval accuracy of different methods    (b) Vanilla retrieval-augmented paradigm of LKE    (c) Our proposed CARML

Figure 1: (a) The trend of retrieval accuracy (Recall@1) for the most relevant editing knowledge as the number of sequential edits increases across different methods. (b) The paradigm of conventional retrieval-augmented lifelong knowledge editing (LKE) methods. (c) Our proposed CARML achieves precise localization of relevant editing knowledge by dynamically integrating multi-channel retrieval information and enables reliable knowledge editing through multi-level collaborative guidance.

Existing knowledge editing research predominantly focuses on large language models (LLMs) domain, primarily addressing single or batch knowledge editing tasks. These approaches can be broadly categorized into two types: (1) Intrinsic knowledge editing [18, 30], which directly modifies specific model parameters to update and store new knowledge; and (2) External knowledge-aided editing [31, 3], which synergizes additional contexts or parametric modules (*e.g.*, in-context learning [45] or Feed-Forward Networks (FFN) [39]) to update knowledge without altering the model itself. While these methods have demonstrated efficacy within LLMs, their direct transposition to MLLMs often encounters substantial resistance. The inclusion of visual modalities and the complex semantic interactions between modalities make it difficult for these approaches to effectively capture cross-modal semantic associations, thereby limiting their capacity to steer MLLMs toward generating desired responses [7, 33].

In practical applications, LLMs or MLLMs often require continuous knowledge updates to reflect the dynamic changes of real-world information, giving rise to the concept of lifelong knowledge editing [2]. Conventional single or batch editing paradigms face acute challenges in such lifelong scenarios. This is primarily the inability to decouple the knowledge associated with different editing requests, which produces the cumulative effect of parameter perturbations, leading to catastrophic forgetting [20]. To counteract this, certain methods construct editable knowledge bases to store each edits independently, and introduce retrieval-augmented techniques to avoid directly modifying the model's original parameters, thereby alleviating forgetting [14, 15]. However, as the volume of editable knowledge grows, these methods encounter increasing difficulties in accurately matching relevant knowledge (Figure 1(a)). More critically, the editable knowledge they architect is generally circumscribed to a single type, lacking the synergistic integration of multi-level knowledge (encompassing unimodal, multimodal, and explicit label information), thus failing to collaboratively guide the editing process. This deficiency fundamentally limits the reliability of the editing outcomes.

To address this critical gap, we introduce CARML, a novel retrieval-augmented editing framework specifically designed to tackle the above challenges. The core innovation of CARML lies in its integration of conflict-aware dynamic retrieval and multi-level implicit and explicit knowledge guidance, enabling reliable lifelong knowledge editing in MLLMs. Its workflow is as follows:

To achieve precise knowledge retrieval, CARML introduces a conflict-aware dynamic retrieval mechanism. By leveraging a pre-trained multimodal embedding model, CARML processes visual, textual and multimodal information of sequential editing requests, constructing knowledge bases tailored to specific modalities. A multi-channel retrieval strategy is then used to query these knowledge bases in parallel for the incoming edit samples, generating modality-specific retrieval candidates. The key innovation lies in CARML's ability to quantify intra-modal uncertainty and inter-modal conflict across retrieval candidates, dynamically weighting and locating the most accurate and consistent retrieved knowledge. To ensure edits are applied only when necessary, CARML constructs an edit scope classifier that maps the embeddings of edit samples and retrieved knowledge into a newly constructed semantic space using two trainable MLPs. By calculating the semantic distance, it

evaluates whether a subsequent editor will be triggered. This mechanism safeguards the locality of editing by effectively filtering out irrelevant edit samples.

To ensure the high reliability of the editing process, CARML incorporates a carefully designed multi-level implicit and explicit knowledge guidance strategy. At the implicit knowledge guidance level, CARML distills retrieved knowledge into information-rich continuous prompt prefixes, which effectively guide MLLMs to generate the desired outcomes. These prefixes encapsulate two innovative types of prompts: (1) Static knowledge prompt, capturing key knowledge semantics in the textual modality. (2) Context-aware dynamic prompts, which dynamically integrate multi-type retrieval knowledge at a fine-grained token level, under the guidance of edit samples. These prompts can provide highly task-specific multimodal instructions tailored to the editing context. At the explicit knowledge guidance level, CARML further enhances the reliability of the editing process by introducing an output logits enhancement mechanism. During inference, this mechanism directly boosts the probabilities of tokens aligned with the true labels in the retrieved knowledge. By decoupling the knowledge update process from the core parameters of the MLLM and selectively integrating the most relevant retrieved knowledge, CARML ensures adherence to three critical editing properties: reliability, generality, and locality.

Our contributions are summarized as follows: **(1)** We architect CARML, a pioneering retrieval-augmented editing framework, specifically conceived for the demands of lifelong knowledge editing in MLLMs. **(2)** We introduce a conflict-aware dynamic retrieval mechanism that uniquely exploits multi-channel retrieval information for accurate knowledge localization based on quantified uncertainty and conflict degree. **(3)** We construct an edit scope classifier, which guarantees the locality of editing by effectively filtering irrelevant edit samples. **(4)** We design an innovative multi-level collaborative guidance strategy, integrating implicit (static and dynamic prompts) and explicit (logits enhancement) knowledge injection to achieve reliable editing.

## 2 Related Works

**Knowledge Editing for LLMs.** Knowledge editing focuses on directly modifying or updating fact-based knowledge or behavior stored within a model without the need for costly full retraining [12, 42]. And an ideal editing method should meet three critical criteria: reliability (accurately modifying the target knowledge), generality (applying the modification to relevant inputs), and locality (preserving unrelated knowledge) [38]. Knowledge editing for LLMs can be broadly categorized into intrinsic knowledge editing and external knowledge-aided methods. The intrinsic knowledge editing approaches involve updating specific model parameters to modify its knowledge. A straightforward strategy is to locate and edit parameters linked to specific knowledge. For instance, ROME [28] uses causal mediation analysis to locate editable regions, while MEMIT [29] supports batch editing via rank-one parameter modifications. Another strategy involves meta-learning, training a hypernetwork to update LLM parameters, as seen in MEND [30] and KE [9]. External knowledge-aided methods aim to store edited knowledge in external memory, leaving the model parameters unchanged. During inference, relevant information is retrieved to revise outputs. Notable examples include SERAC [31], IKE [45], and RECIPE [3].

**Knowledge Editing for MLLMs.** Applying knowledge editing to multimodal large language models (MLLMs) is an emerging but challenging field [40, 32, 26]. Due to the complexity of multimodal information, directly adopting LLM editing methods often yields suboptimal performance [11, 4]. Research on MLLM knowledge editing is still in its early stages, with only a few methods tailored to multimodal scenarios. For instance, UniKE [33] proposes a unified framework that combines intrinsic knowledge editing with external memory assistance. LiveEdit [2] generates independent low-rank experts for each editing instance, employing hard routing and soft routing mechanisms to select and combine these experts. Despite these advancements, achieving reliable and continuous knowledge editing for MLLMs, particularly for lifelong knowledge editing that supports sustained updates, remains an open and pressing challenge.

## 3 Method

In this section, we first formalize the problem of the lifelong multimodal knowledge editing (Section 3.1). We then provide an overview of our proposed CARML framework (Section 3.2), followed by

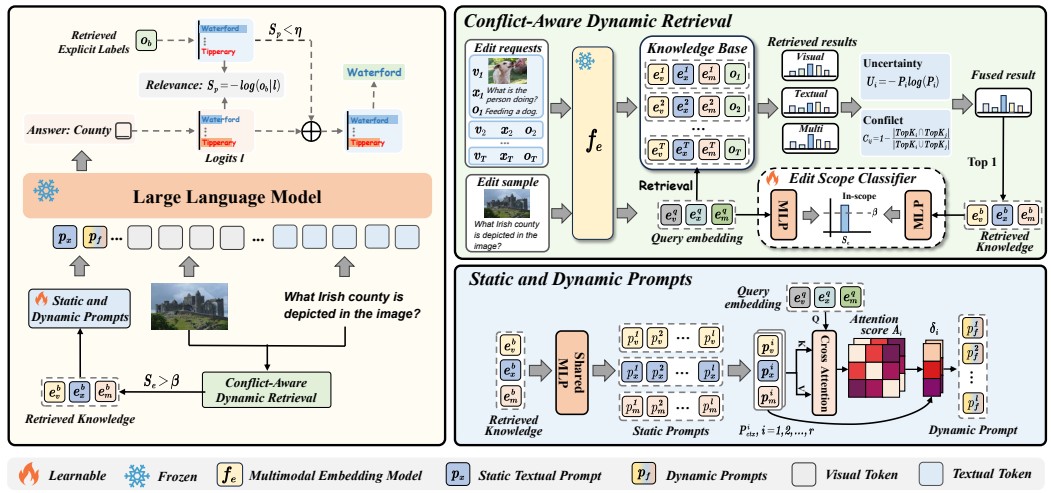

Figure 2: The overall architecture of the proposed CARML. It consists of two key modules: (1) Conflict-aware dynamic retrieval module: locates the most relevant knowledge for the edit sample and determines whether to initiate the subsequent editing process. (2) Multi-level knowledge guidance: delivers both implicit guidance (static and dynamic prompts) and explicit guidance (logits enhancement) for reliable editing of MLLM.

detailed descriptions of its two core modules: Conflict-Aware Dynamic Retrieval (Section 3.3) and Multi-Level Knowledge Guidance (Section 3.4).

## 3.1 Background: Lifelong Multimodal Knowledge Editing

We focus on the task of lifelong multimodal knowledge editing. Given a Multimodal Large Language Model (MLLM) $f_\Theta : v \times x \to o$, parameterized by $\Theta$, which maps an input image $v$ and text $x$ to a prediction $o = f_\Theta(v, x)$. The editing process involves a sequence of edit requests evolving over time, denoted as $D_{edit} = \{(v_t, x_t, o_t)\}_{t=1}^{T}$, where $(v_t, x_t)$ is the target input requiring editing, and $o_t$ is the desired output. At time step $T$, a Model Editor (ME) receives the model $f_{\Theta_{T-1}}$ (edited at step $T-1$), along with the current edit request $(v_T, x_T, o_T)$. It subsequently yields an updated model $f_{\Theta_T}$ as follows:

$$f_{\Theta_T} = \text{ME}\left(f_{\Theta_{T-1}}, v_T, x_T, o_T\right), \quad \text{s.t.} \ f_{\Theta_T}(v, x) = \begin{cases} o_t & \text{if } (v_t, x_t) \in D_{edit}, \\ f_\Theta(v, x) & \text{if } (v, x) \notin D_{edit}. \end{cases} \quad (1)$$

This equation stipulates that subsequent to an edit, the updated MLLM $f_{\Theta_T}$ should correctly predict the outcome for the current edit request, *i.e.*, $f_{\Theta_T}(v_T, x_T) = o_T$, while maintaining effective edits for past knowledge $(v_t, x_t) \in D_{edit}$. Additionally, for unrelated data $(v, x) \notin D_{edit}$, the model should retain the original capabilities of $f_\Theta$.

## 3.2 Overview of CARML

CARML is a novel retrieval-augmented editing framework designed for reliable lifelong multimodal knowledge editing, as shown in Figure 2. Its core workflow is as follows: Initially, CARML constructs an evolving multi-type edit knowledge base that stores historical edits context. Upon the arrival of a new edit sample, CARML instigates its innovative Conflict-Aware Dynamic Retrieval mechanism. This module utilizes a multi-channel retrieval strategy and quantifies intra-modal uncertainty and inter-modal conflicts, thereby adaptively fusing multi-channel retrieval information and re-ranking candidates to ensure highly accurate retrieval. Then, an edit scope classifier discerns whether the edit sample semantically aligns with the editing scope of the retrieved knowledge, a crucial step

for preserving model locality. If deemed in-scope, CARML activates a synergistic Multi-Level Knowledge Guidance strategy, comprising implicit guidance (static and dynamic prompts) and explicit guidance (logits enhancement). These strategies collaboratively navigate the MLLM towards generating outputs that faithfully adhere to the edit directive.

## 3.3 Conflict-Aware Dynamic Retrieval

To achieve precise retrieval within a continuously growing knowledge base, CARML introduces a conflict-aware dynamic retrieval mechanism comprising four key components: knowledge base construction, multi-channel retrieval, conflict-aware fusion and re-ranking, and an edit scope classifier.

**Knowledge Base Construction.** CARML first curates a multi-type edit knowledge base $K = \{k_t = (e_v^t, e_x^t, e_m^t, o_t)\}_{t=1}^T$, where each entry $k_t$ corresponds to a historical edit request $(v_t, x_t, o_t)$. The knowledge base is initialized as empty and grows with new edit requests. The embeddings $e_v^t, e_x^t, e_m^t \in \mathbb{R}^d$ represent the visual, textual, and fused multimodal embeddings, respectively, extracted using a frozen, pre-trained multimodal embedding model $f_E$ (*e.g.*, mexma-siglip2 [36]):

$$e_v^t = f_E^v(v_t), e_x^t = f_E^x(x_t + o_t), e_m^t = LayerNorm(e_v^t + e_x^t) \tag{2}$$

where $f_E^v$ and $f_E^x$ denote the visual and textual encoders of the embedding model $f_E$, and $(x_t + o_t)$ stands for the string concatenation of $x_t$ and $o_t$.

**Multi-Channel Retrieval.** Given an incoming edit sample $q = (v_q, x_q)$, its corresponding query embeddings $q_e = (e_v^q, e_x^q, e_m^q)$ are computed via the similar process outlined in Equation 3.3. Subsequently, CARML executes a multi-channel retrieval strategy to identify relevant entries from the knowledge base $K$. This strategy computes the cosine similarity score between the query embedding $q_e$ and each modality-specific embedding sets within the knowledge base $K$.

$$S_v = sim\left(e_v^q, \{e_v^t\}_{t=1}^T\right), S_x = sim\left(e_x^q, \{e_x^t\}_{t=1}^T\right), S_m = sim\left(e_m^q, \{e_m^t\}_{t=1}^T\right) \tag{3}$$

where $sim(\cdot, \cdot)$ denotes the cosine similarity function. The sets $\{e_v^t\}_{t=1}^T$, $\{e_x^t\}_{t=1}^T$ and $\{e_m^t\}_{t=1}^T$ represent the visual, textual and multimodal embedding collections within the knowledge base $K$, respectively. And $S_v, S_x, S_m$ are the resultant modality-specific similarity score vectors.

**Conflict-Aware Fusion and Re-ranking.** To judiciously integrate information from multi-channels and manage potential inter-modal discrepancies, we devise a conflict-aware fusion and re-ranking mechanism. The procedure is as follows:

For the similarity score $S_i$ of each modality $i \in \{v, x, m\}$, we first convert it into probability distribution $P_i = softmax(S_i)$. The information entropy $U_i = -P_i \log(P_i)$ is then calculated for each distribution, quantifying the internal uncertainty of the retrieval results for modality $i$.

Beyond intra-modal uncertainty, the degree of decisional conflict among modalities is paramount for assessing fusion reliability. We first identify the index set $TopK_i$ of the top-K candidates exhibiting the highest similarity scores for each modality $i$. For any pair of distinct modalities $i$ and $j$, the Jaccard similarity coefficient $J_{ij}$ of their $TopK_i$ and $TopK_j$ index sets is computed. Then, the conflict degree $C_{ij}$ between modalities $i$ and $j$ is defined as:

$$C_{ij} = 1 - \frac{|TopK_i \cap TopK_j|}{|TopK_i \cup TopK_j|} \tag{4}$$

Considering both intra-modal uncertainty $U_i$ and inter-modal conflicts $C_{ij}$, a dynamic weighting mechanism is introduced to adaptively assign a weight $W_i$ to each modality. This weight aims to reward modalities that are both low-uncertainty and less conflict with others. Specifically, we first calculate an intermediate score $m_i$ for each modality:

$$m_i = \frac{1}{2} \cdot \left(1 - \frac{U_i}{\sum_k U_k}\right) + \frac{1}{2} \cdot \left(1 - \frac{\sum_{i \neq j} C_{ij}}{\sum_{k,l; k \neq l} C_{kl}}\right) \tag{5}$$

where the first term denotes the normalized entropy, the second term represents normalized conflict degree for modality $i$. And the final dynamic weights are procured via $W_i = softmax(m_i)$.

Finally, these dynamic weights $W_i$ are employed to perform a weighted sum of the original similarity scores $S_i$ for each modality, yielding the definitive fused similarity score vector $S_f = \sum_{i \in \{v,x,m\}} W_i \cdot S_i$. Then, all candidates in the knowledge base are re-ranked based on $S_f$, and the single premier entry exhibiting the highest fused score is selected as the final retrieved knowledge $k_b = \left(e_v^b, e_x^b, e_m^b, o_b\right)$.

**Edit Scope Classifier.** To stringently preserve model locality, CARML integrates an edit scope classifier. This component determines whether the edit sample $q$ falls within the editing scope of the retrieved knowledge $k_b$.

Specifically, two lightweight, trainable MLPs are used to map the current edit sample's query embedding $q_e$ and the retrieved knowledge $k_b$ into a fresh semantic spaces. These MLPs are trained with the objective of minimizing the distance between in-scope query-knowledge pairs and maximizing it for out-of-scope pairs in this projected space. The mean cosine similarity $S_e$ between the mapped query and retrieved knowledge embeddings is then computed. If $S_e$ exceeds a predefined threshold $\beta$, the current edit sample $q$ is classified as in-scope; otherwise, it is deemed out-of-scope.

This scope classifier is optimized using the Focal Loss $\mathcal{L}_{scope}$ [23], enabling it to concentrate on hard-to-classify instances. Ground-truth labels for training data are predefined: reliability and generality samples (which should be affected by the edit) are labeled as "in-scope", while locality samples (which must remain unaffected) are designated as "out-of-scope". The loss function is:

$$\mathcal{L}_{scope} = -y(1-\hat{y})^2 \log\left(\hat{y}\right) - (1-y)\left(\hat{y}\right)^2 \log\left(1 - \hat{y}\right) \tag{6}$$

where $y$ is the ground-truth label, $\hat{y}$ is the classifier's predicted probability.

## 3.4 Multi-Level Knowledge Guidance

When the edit sample $q$ is classified as in-scope, CARML activates its multi-level knowledge guidance module. This module leverages the retrieved knowledge $k_b = \left(e_v^b, e_x^b, e_m^b, o_b\right)$ to provide both implicit and explicit guidance to the MLLM, steering it towards generating the desired output $o_b$.

### 3.4.1 Implicit Guidance: Static and Dynamic Prompts

Inspired by prompt tuning [25, 46], we distill the retrieved knowledge $k_b$ into parameter-efficient continuous prompt, serving as implicit reference input to the MLLM. These continuous prompts are prepended to the MLLM's original input embedding sequence and comprise two distinct types:

**Static Knowledge Prompt.** This prompt is engineered to encapsulate the core semantic essence of the textual knowledge statement contained within the retrieved $k_b$. A lightweight MLP ($MLP_p$) is utilized to map the retrieved knowledge's text embedding $e_x^b$ to a fixed-length continuous prompt $p_x$:

$$p_x = f_{resp}(MLP_p(e_x^b)) \tag{7}$$

where $p_x \in \mathbb{R}^{r \times d^{mllm}}$, $r$ is the length of the continuous prompt, $d_{mllm}$ is the word embedding dimension of the MLLM, and $f_{resp}$ maps the MLP output vector into the desired matrix shape.

**Context-Aware Dynamic Prompt.** Since static knowledge prompts may lack the specificity required for intricate multimodal edits, we introduce the context-aware dynamic prompt. This generates customized multimodal instructions that are highly attuned to the interaction between the edit sample $q$ and the retrieved knowledge $k_b$. Initially, the visual ($e_v^b$) and multimodal ($e_m^b$) embeddings of the retrieved knowledge $k_b$ are transformed via shared-weight $MLP_p$ (Equation 3.4.1) to produce continuous prompts $p_v$ and $p_m$ of length $r$, respectively. Then, we propose a token-level attention fusion mechanism to further explore the fine-grained associations between the edit sample $q$ and retrieved knowledge $k_b$. For each token position $i$ (from 1 to $r$), the edit sample embedding $q_e$ ($e_v^q, e_x^q, e_m^q$, forming as a $3 \times d^{mllm}$ matrix $Q_e$) serves as the query, while the $i$-th tokens from the three modality prompts ($p_x^i, p_v^i, p_m^i$, forming a $3 \times d^{mllm}$ matrix $P_{ctx}^i$), serve as keys and values.

This calculates a token-specific attention score $A_i \in \mathbb{R}^{3 \times 3}$:

$$A_i = softmax\left(\frac{Q_e \cdot (P_{ctx}^i)^T}{\sqrt{d}}\right), i = 1, 2, ..., r \qquad (8)$$

To derive context-aware fused prompt token $p_f^i$, we first sum the columns of $A_i$ to obtain a $1 \times 3$ vector $A_i'$ representing the aggregated importance of each prompt modality $p_x^i, p_v^i, p_m^i$ with respect to the edit sample query $q_e$. Later, we set the fusion weight $\delta_i = softmax(A_i')$. The $i$-th fine-grained fused prompt token $p_f^i$ is then computed as a weighted sum: $p_f^i = P_{ctx}^i \cdot \delta_i$. Repeating this fusion for all $i = 1, 2, ..., r$, we can obtain the final context-aware dynamic prompt $p_f \in \mathbb{R}^{r \times d^{mllm}}$.

Finally, the static knowledge prompt $p_x$, context-aware dynamic prompt $p_f$, and the MLLM embeddings for the query image $f_{emb}(v_q)$ and text $f_{emb}(x_q)$ are concatenated to form the input sequence of MLLM. The inference process is reformulated as: $o = f_\Theta(p_x \oplus p_f \oplus f_{emb}(v_q) \oplus f_{emb}(x_q))$.

### 3.4.2 Explicit Guidance: Output Logits Enhancement

To further enhance editing reliability, we introduce an explicit token bias mechanism that directly adjusts the MLLM's output logits. Specifically, during inference, let $l \in \mathbb{R}^V$ be the initial probability distribution predicted by the MLLM after applying softmax, where $V$ is the vocabulary size, and $o_b$ is the ideal output token ids from the retrieved knowledge $k_b$. Therefore, thanks to the high precision of the retrieval strategy described above, we update the logits distribution $l$ simply by:

$$l' = l + h(o_b), \ if \ S_p = -log(o_b|l) < \eta \qquad (9)$$

Where $h(o_b) \in \mathbb{R}^V$ corresponds to the one-hot encoding of the desired token id $o_b$. The similarity $S_p$ is used to measure the relevance between the original probability distribution $l$ and the desired output $o_b$. Only when it is below the threshold $\eta$, the corresponding logits enhancement strategy is initiated, which further protects the model's original ability to irrelevant inputs.

### 3.5 Training Objective

The training objective of CARML is to jointly optimize the scope classifier and the implicit guidance modules, while keeping the core MLLM parameters frozen. Following previous work [2], we define the editing loss $\mathcal{L}_{edit}$ to ensure that edits satisfy reliability, generality, and locality requirements. And the overall training loss $\mathcal{L}_{total}$ is as follows:

$$\mathcal{L}_{total} = \mathcal{L}_{edit} + \lambda \cdot \mathcal{L}_{scope} \qquad (10)$$

where $\lambda$ is a weight balancing the two losses.

## 4 Experiments

### 4.1 Experimental Setup

**Datasets and Models.** Following [7], our experiments are conducted on the MMEdit benchmark [7], which includes two sub-tasks: Editing VQA (E-VQA) and Editing Image Caption (E-IC). Additionally, we incorporated VLKEB [17] dataset, which consists of real images to better represent real-world scenarios. Besides, experiments are executed on two prominent MLLMs: BLIP 2-OPT (2.7B) [21] and LLaVA-V1.5 (7B) [24].

**Baselines.** CARML's performance is compared with the following baselines: (1) FT-L [7]: direct fine-tuning of the last layer of the LLM. (2) Two intrinsic knowledge editing methods: TP [18] and MEND [30]. (3) External knowledge-aided methods: SERAC [31], LEMoE [39], and RECIPE [3], along with the state-of-the-art lifelong editing method LiveEdit [2], designed specifically for MLLM.

**Implementation Details.** Across all experiments, we utilize mexma-siglip2 [36] as the multimodal embedding model. Within CARML's conflict-aware dynamic retrieval module, we select the top 15 retrieval candidates to calculate inter-modal conflict degree. For the multi-level knowledge guidance module, the length $r$ of each type of prompt is set to 4. And the weight coefficient $\lambda$ in the final loss function is set to 0.04. More details of the experimental setup are shown in Appendix A.

Table 1: Main editing results for E-VQA dataset. $T$: Num Edits.

| Method | E-VQA | | | | | | | | | | | | | | |
|---|---|---|---|---|---|---|---|---|---|---|---|---|---|---|---|
| | $T=1$ | | | | | $T=100$ | | | | | $T=1000$ | | | | |
| | Rel. | T-Gen. | M-Gen. | T-Loc. | M-Loc. | Rel. | T-Gen. | M-Gen. | T-Loc. | M-Loc. | Rel. | T-Gen. | M-Gen. | T-Loc. | M-Loc. |
| BLIP-2 OPT | | | | | | | | | | | | | | | |
| FT-L | 64.6 | 57.5 | 41.9 | 91.7 | 84.5 | 53.5 | 47.6 | 49.3 | 75.2 | 37.4 | 38.3 | 32.3 | 37.5 | 41.6 | 28.6 |
| TP | 70.1 | 65.8 | 53.1 | 98.1 | 85.3 | 44.3 | 38.2 | 33.3 | 43.8 | 38.5 | 20.6 | 15.1 | 18.4 | 8.7 | 8.3 |
| MEND | 93.1 | 92.8 | 93.1 | 92.0 | 75.8 | 17.7 | 16.4 | 18.3 | 91.5 | 67.9 | 15.8 | 14.4 | 17.7 | 91.7 | 70.2 |
| SERAC | 88.4 | 84.5 | 84.3 | 85.8 | 26.0 | 87.0 | 81.3 | 81.0 | 71.0 | 15.6 | 83.4 | 70.8 | 80.3 | 67.7 | 13.1 |
| LEMoE | 93.6 | 92.2 | 91.4 | 98.5 | 85.2 | 29.5 | 22.6 | 28.4 | 81.7 | 22.7 | 20.5 | 15.0 | 20.2 | 73.5 | 22.6 |
| RECIPE | 87.4 | 85.0 | 86.6 | 99.9 | 88.4 | 85.2 | 82.7 | 84.2 | 98.7 | 83.9 | 82.6 | 74.3 | 79.5 | 97.1 | 81.5 |
| LiveEdit | 96.7 | 94.2 | 93.8 | 100.0 | **100.0** | 95.3 | 92.9 | 85.5 | 100.0 | **99.8** | 94.4 | 92.0 | 84.7 | 100.0 | **97.4** |
| **CARML** | **100.0** | **100.0** | **100.0** | **100.0** | 97.0 | **97.3** | **96.7** | **94.9** | **100.0** | 97.7 | **97.0** | **95.9** | **93.2** | **100.0** | 91.9 |
| LLaVA-V1.5 | | | | | | | | | | | | | | | |
| FT-L | 96.4 | 92.7 | 93.4 | 76.5 | 72.1 | 68.8 | 60.1 | 62.6 | 42.8 | 34.6 | 44.2 | 38.2 | 41.6 | 33.6 | 24.7 |
| TP | 36.0 | 36.1 | 28.7 | 93.9 | 97.6 | 29.4 | 28.7 | 24.7 | 14.6 | 45.0 | 16.6 | 16.8 | 15.7 | 7.3 | 15.6 |
| MEND | 91.2 | 90.0 | 91.3 | 91.0 | 90.2 | 2.2 | 2.2 | 2.2 | 0.2 | 0.6 | 0.0 | 0.1 | 0.1 | 0.1 | 0.1 |
| SERAC | 89.3 | 83.7 | 85.0 | 82.0 | 23.8 | 88.1 | 81.5 | 82.5 | 62.1 | 12.9 | 85.6 | 75.6 | 82.0 | 62.5 | 15.7 |
| LEMoE | 93.6 | 92.8 | 90.0 | 99.3 | 97.0 | 43.0 | 37.2 | 34.6 | 78.1 | 50.4 | 31.4 | 25.3 | 27.4 | 70.1 | 41.5 |
| RECIPE | 91.8 | 87.1 | 87.2 | 95.1 | 87.7 | 88.3 | 81.6 | 82.8 | 89.4 | 81.2 | 83.5 | 72.1 | 81.6 | 84.0 | 80.3 |
| LiveEdit | 94.3 | 94.5 | 88.0 | 100.0 | **100.0** | 93.5 | 92.3 | 85.9 | 100.0 | **99.3** | 92.9 | 90.2 | 84.3 | 100.0 | **96.4** |
| **CARML** | **100.0** | **99.7** | **98.8** | **100.0** | 99.9 | **100.0** | **99.5** | **98.8** | **100.0** | 96.6 | **97.1** | **95.9** | **92.8** | **100.0** | 94.9 |

## 4.2 Main Results

**Competitive Performance of CARML.** Tables 1, 2 and 3 present the results of lifelong editing experiments conducted on the MMEdit and VLKEB datasets for CARML and baseline methods. The experimental reveal the following findings: While most conventional methods can achieve satisfactory reliability in single-edit scenarios, their performance significantly degrades as the number of editing steps ($T$) increases. This is because, as the volume of edited knowledge grows, fine-tuning methods like FT-L and intrinsic knowledge editing approaches such as TP and MEND suffer from catastrophic forgetting due to accumulated parameter shifts. On the other hand, external knowledge-aided methods like LEMoE, RECIPE, and LiveEdit increasingly struggle to accurately retrieve relevant knowledge from an expanding knowledge base. In contrast, our proposed CARML consistently maintains nearly 100% performance even when the number of edits reaches 1000. This remarkable performance is primarily attributed to its ingenious conflict-aware dynamic retrieval strategy, which precisely locates the most relevant knowledge for each edit sample. Moreover, CARML's multi-level knowledge guidance mechanism robustly steers the MLLM at both implicit and explicit levels to generate the desired outputs.

Table 2: Main editing results for E-IC dataset. $T$: Num Edits.

| Method | E-IC | | | | | | | | | | | | | | |
|---|---|---|---|---|---|---|---|---|---|---|---|---|---|---|---|
| | $T=1$ | | | | | $T=100$ | | | | | $T=1000$ | | | | |
| | Rel. | T-Gen. | M-Gen. | T-Loc. | M-Loc. | Rel. | T-Gen. | M-Gen. | T-Loc. | M-Loc. | Rel. | T-Gen. | M-Gen. | T-Loc. | M-Loc. |
| BLIP-2 OPT | | | | | | | | | | | | | | | |
| FT-L | 40.6 | 41.2 | 38.7 | 97.7 | 98.9 | 44.5 | 45.1 | 39.4 | 83.2 | 53.4 | 45.8 | 42.2 | 43.7 | 52.6 | 54.1 |
| TP | 49.7 | 48.6 | 46.0 | 93.7 | 79.0 | 37.5 | 37.6 | 33.8 | 10.3 | 20.9 | 26.0 | 26.3 | 24.9 | 4.1 | 11.8 |
| MEND | 95.0 | 92.5 | 92.3 | 95.0 | 8.9 | 8.0 | 8.2 | 8.0 | 20.2 | 23.2 | 6.5 | 6.5 | 6.5 | 13.5 | 20.4 |
| SERAC | 88.7 | 83.8 | 84.4 | 84.3 | 24.7 | 43.6 | 42.2 | 39.1 | 57.8 | 15.8 | 43.1 | 41.7 | 38.7 | 48.1 | 14.9 |
| LEMoE | 93.1 | 91.4 | 83.3 | 94.5 | 60.4 | 55.9 | 52.3 | 49.2 | 92.1 | 53.7 | 41.6 | 40.9 | 40.2 | 93.8 | 65.5 |
| RECIPE | 80.7 | 79.2 | 78.9 | 100.0 | 95.3 | 41.5 | 40.2 | 40.6 | 98.8 | 93.5 | 37.0 | 38.6 | 37.2 | 99.8 | 92.7 |
| LiveEdit | 80.6 | 80.1 | 76.9 | 100.0 | 100.0 | 79.2 | 77.4 | 74.1 | 100.0 | 100.0 | 72.9 | 70.3 | 67.9 | 100.0 | 100.0 |
| **CARML** | **99.4** | **99.5** | **99.6** | **100.0** | **100.0** | **99.4** | **99.5** | **99.5** | **100.0** | **100.0** | **99.4** | **99.5** | **99.3** | **100.0** | **100.0** |
| LLaVA-V1.5 | | | | | | | | | | | | | | | |
| FT-L | 72.1 | 71.9 | 66.1 | 99.1 | 98.7 | 64.2 | 56.5 | 53.2 | 85.5 | 91.8 | 58.0 | 52.3 | 53.8 | 65.2 | 80.3 |
| TP | 57.6 | 59.2 | 55.3 | 60.9 | 88.0 | 22.9 | 25.8 | 20.9 | 3.6 | 14.9 | 10.3 | 13.1 | 9.8 | 1.7 | 4.5 |
| MEND | 92.8 | 91.8 | 90.6 | 96.4 | 93.7 | 56.8 | 56.9 | 53.1 | 87.6 | 84.6 | 54.4 | 54.1 | 51.0 | 83.9 | 80.6 |
| SERAC | 88.2 | 81.0 | 85.6 | 84.0 | 28.6 | 53.4 | 53.7 | 49.4 | 48.0 | 17.3 | 52.9 | 53.4 | 49.0 | 49.9 | 16.7 |
| LEMoE | 93.8 | 91.4 | 90.6 | 95.1 | 93.0 | 56.7 | 53.2 | 52.0 | 92.7 | 79.9 | 36.7 | 33.1 | 29.7 | 85.2 | 77.1 |
| RECIPE | 85.9 | 76.1 | 83.3 | 96.7 | 95.2 | 55.3 | 53.2 | 51.9 | 91.3 | 92.5 | 51.7 | 53.4 | 48.5 | 88.9 | 91.4 |
| LiveEdit | 82.2 | 81.0 | 78.3 | 100.0 | 100.0 | 80.8 | 78.8 | 63.5 | 100.0 | 100.0 | 72.8 | 70.0 | 57.1 | 100.0 | 99.8 |
| **CARML** | **99.9** | **99.8** | **99.8** | **100.0** | **100.0** | **99.9** | **99.8** | **99.8** | **100.0** | **100.0** | **99.3** | **99.4** | **99.2** | **100.0** | **100.0** |

**Cross-task Editing Evaluation.** In the process of cross-task editing, MLLMs face the challenging task of simultaneously handling and editing samples from different tasks (E-VQA and E-IC) in a continuous editing sequence. Table 4 summarizes the results for 1000 sequential edits (the average of E-VQA and E-IC test results) based on the BLIP 2-OPT model in this demanding scenario. Evidently, most baseline methods struggle to effectively edit both tasks within a single editing sequence.

Table 3: Main editing results for VLKEB dataset. $T$: Num Edits.

| Method | VLKEB | | | | | | | | | | | | | | |
|---|---|---|---|---|---|---|---|---|---|---|---|---|---|---|---|
| | $T = 1$ | | | | | $T = 100$ | | | | | $T = 1000$ | | | | |
| | Rel. | T-Gen. | M-Gen. | T-Loc. | M-Loc. | Rel. | T-Gen. | M-Gen. | T-Loc. | M-Loc. | Rel. | T-Gen. | M-Gen. | T-Loc. | M-Loc. |
| BLIP-2 OPT | | | | | | | | | | | | | | | |
| FT-L | 54.8 | 54.1 | 55.2 | 98.7 | 95.1 | 56.4 | 57.5 | 56.7 | 86.4 | 70.2 | 55.9 | 55.7 | 55.1 | 52.4 | 52.7 |
| TP | 51.0 | 49.5 | 50.9 | 94.8 | 78.6 | 46.6 | 48.1 | 47.2 | 64.3 | 43.2 | 24.4 | 24.2 | 24.3 | 16.4 | 20.0 |
| MEND | 94.9 | 93.8 | 93.8 | 95.0 | 86.5 | 39.9 | 41.0 | 40.4 | 92.3 | 84.3 | 37.2 | 38.0 | 37.2 | 91.5 | 84.1 |
| SERAC | 88.0 | 84.7 | 85.2 | 68.1 | 17.8 | 66.7 | 53.7 | 64.4 | 59.2 | 17.9 | 53.6 | 45.8 | 52.4 | 56.8 | 16.9 |
| LEMoE | 94.6 | 93.1 | 92.4 | 94.5 | 61.5 | 47.7 | 48.3 | 47.5 | 52.8 | 51.5 | 37.7 | 37.2 | 38.7 | 52.7 | 55.5 |
| RECIPE | 93.1 | 91.5 | 88.0 | 97.7 | 93.5 | 63.5 | 58.2 | 64.0 | 93.5 | 94.2 | 51.8 | 48.2 | 50.6 | 91.0 | 92.8 |
| LiveEdit | 98.8 | 98.1 | 94.9 | 100.0 | **100.0** | 98.2 | 97.7 | 94.0 | 100.0 | **100.0** | 97.0 | 91.9 | 87.5 | 100.0 | **100.0** |
| **CARML** | **99.1** | **99.2** | **99.1** | **100.0** | 95.5 | **99.1** | **99.2** | **98.9** | **100.0** | 95.5 | **99.1** | **99.1** | **98.8** | **100.0** | 95.5 |
| LLaVA-V1.5 | | | | | | | | | | | | | | | |
| FT-L | 92.8 | 89.1 | 92.6 | 92.1 | 91.8 | 74.1 | 74.8 | 75.6 | 68.7 | 83.4 | 68.4 | 67.9 | 67.2 | 66.8 | 74.9 |
| TP | 50.8 | 55.7 | 51.7 | 87.9 | 90.4 | 19.7 | 20.1 | 19.4 | 11.4 | 24.1 | 5.5 | 4.8 | 5.5 | 2.8 | 7.2 |
| MEND | 92.1 | 91.3 | 90.2 | 89.2 | 90.1 | 0.6 | 0.6 | 0.7 | 0.2 | 0.1 | 0.0 | 0.1 | 0.1 | 0.1 | 0.1 |
| SERAC | 90.0 | 89.1 | 87.9 | 66.7 | 14.2 | 72.3 | 62.4 | 70.7 | 53.7 | 13.7 | 60.9 | 56.5 | 60.1 | 52.9 | 15.0 |
| LEMoE | 94.9 | 93.1 | 91.7 | 87.0 | 87.9 | 78.3 | 76.2 | 73.5 | 50.9 | 49.2 | 63.2 | 57.5 | 56.8 | 48.4 | 45.6 |
| RECIPE | 93.1 | 92.8 | 92.1 | 91.7 | 86.4 | 77.2 | 66.0 | 76.4 | 88.5 | 84.1 | 63.8 | 56.2 | 61.9 | 86.7 | 81.2 |
| LiveEdit | 96.4 | 95.2 | 93.7 | 100.0 | **100.0** | 94.6 | 90.7 | 89.6 | 100.0 | **100.0** | 92.2 | 84.0 | 82.8 | 100.0 | **100.0** |
| **CARML** | **99.4** | **99.4** | **99.3** | **100.0** | 95.0 | **99.4** | **99.4** | **99.2** | **100.0** | 95.3 | **99.4** | **99.3** | **99.0** | **100.0** | 95.3 |

Table 4: Main results on cross-task editing.

| Method | Rel. | T-Gen. | M-Gen. | T-Loc. | M-Loc. |
|---|---|---|---|---|---|
| FT-L | 29.7 | 25.0 | 28.9 | 2.6 | 6.3 |
| LEMoE | 22.4 | 20.8 | 21.5 | 70.3 | 16.3 |
| RECIPE | 81.2 | 77.7 | 78.8 | 99.9 | 65.9 |
| LiveEdit | 89.0 | 87.6 | 69.4 | 100.0 | **99.3** |
| **CARML** | **94.5** | **94.2** | **92.8** | **100.0** | 92.1 |

Table 5: Ablation study of CARML.

| Method | Rel. | T-Gen. | M-Gen. | T-Loc. | M-Loc. |
|---|---|---|---|---|---|
| *Base* | 37.0 | 38.6 | 37.2 | 99.8 | 92.7 |
| *+Retrieval* | 71.4 | 71.9 | 70.6 | 99.1 | 92.5 |
| *+Classifier* | 70.2 | 73.4 | 69.4 | 100.0 | 100.0 |
| *+C-prompt* | 83.4 | 81.0 | 82.7 | 100.0 | 100.0 |
| *+Logits* | **99.4** | **99.5** | **99.3** | **100.0** | **100.0** |

However, CARML demonstrates remarkable capability in integrating and editing knowledge from these diverse tasks. It outperforms all baseline methods in key metrics such as reliability and generality, highlighting its adaptability and dependable knowledge editing capability.

### 4.3 Further Analysis

**Effect of Individual Components.** To analyze the efficacy of the core components within CARML, we conduct a detailed ablation study with results presented in Table 5. The experiment takes RECIPE [3] as the *base* model and is carried out in 1000 editing scenarios on the E-IC dataset. First, we replace RECIPE's unimodal text retrieval with our conflict-aware dynamic retrieval strategy (*+Retrieval*), resulting in a significant performance boost due to improved retrieval accuracy. This highlights the importance of precise knowledge matching in enhancing editing performance. Next, we substitute RECIPE's original classification strategy with our edit scope classifier (*+Classifier*), which significantly improved locality by better identifying out-of-scope edit samples. We then incorporate context-aware dynamic prompts (*+C-prompt*) into the model's prompt prefix further enhanced reliability and generality. This improvement stems from capturing fine-grained cross-modal correlations between the editing sample and retrieved knowledge, enabling tailored multimodal instructions to guide the model effectively. Lastly, adding the explicit output logits enhancement mechanism (*+Logits*) further strengthened the model's reliability and generality.

**Analysis of the Semantic Space for Edit Scope Classifier.** We employ t-SNE [35] for dimensionality reduction and visualized the embedding distribution of retrieved knowledge (blue) and out-of-scope edit samples (red) within the edit scope classifier, as shown in Figure 3.(a). This provides an intuitive confirmation of the effectiveness of the edit scope classifier: it successfully maximizes the distance between out-of-scope samples and retrieved knowledge within the learned semantic space. Such clear semantic distinction is critical for CARML to perform reliable edits, preventing unintended modifications to unrelated knowledge.

**Analysis of the Trade-off between Accuracy and Editing Time.** Figure 3.(b) illustrates the trade-off between accuracy and overall editing time (including single-edit latency and inference delay after sequential edits) across different methods. It can be observed that CARML achieves an outstanding balance of high accuracy and low time cost, outperforming other baseline methods. This not only

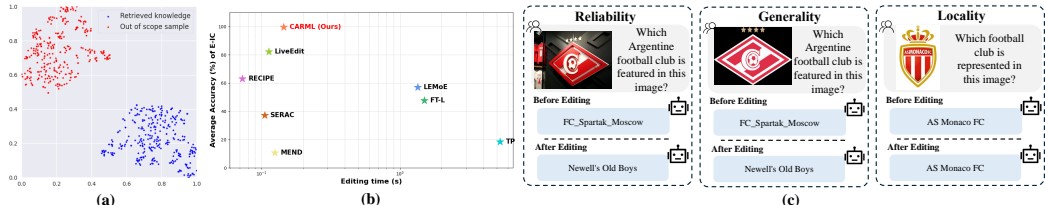

Figure 3: (a) Visualization of the semantic space of retrieved knowledge and out-of-scope samples. (b) The trade-off between accuracy and editing time on the E-IC dataset, where editing time is the sum of single edits and the time taken for model inference after 1000 edits. (c) Visualization example.

highlights CARML's exceptional efficiency-to-performance ratio but also demonstrates its practical applicability and scalability in demanding lifelong multimodal editing scenarios.

**Visualization Examples.** As shown in Figure 3.c and Appendix E, CARML achieves reliable lifelong multimodal editing, which enables precise editing of relevant edit samples and maintains the original knowledge of irrelevant samples.

## 5 Conclusion

To address the challenges of lifelong knowledge editing for MLLMs, this paper proposes a novel retrieval-augmented editing framework named CARML. Its core innovation lies in the organic integration of the conflict-aware dynamic retrieval mechanism and multi-level knowledge guidance strategy. The former dynamically fuses multi-channel retrieval information and incorporates an edit scope classifier, to achieve precise localization and scope determination of the edited knowledge. The latter employs implicit static and dynamic prompts, along with explicit output logit enhancement, to collaboratively guide the model in generating desired outcomes. Extensive experiments demonstrate that CARML significantly outperforms existing methods in lifelong multimodal editing scenarios.

## 6 Limitations

Despite the promising results, the proposed CARML has several limitations: (**1**) Generalization in Complex Editing: While CARML exhibits strong performance in tasks such as VQA and image caption editing, its capability to handle highly complex and abstract editing (*e.g.* intricate common-sense reasoning) requires further investigation. (**2**) Model Scale Constraints: Due to computational limitations, our evaluation is currently restricted to MLLMs with a specific parameter scale. The adaptability and performance of CARML on larger-scale models (*e.g.*, InternVL2.5-78B [6]) remain to be explored.

## Acknowledgment

This work was supported by the National Natural Science Foundation of China (NSFC) under Grant 62476260, 62225207 and 62436008, the Fundamental Research Funds for the Central Universities under Grant WK2100000057.

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

# Appendix

## A  Detailed Experimental Settings

### A.1  Description of Datasets

To evaluate the effectiveness of our proposed CARML, we conduct experiments on three multimodal knowledge editing datasets: E-VQA [7], E-IC [7], and VLKEB [17]. E-VQA and E-IC are part of the MMEdit benchmark, while VLKEB includes real-world images to simulate practical scenarios. Regarding the data split of all three datasets, we followed the setup in the original dataset. The details of each dataset are as follows:

**E-VQA** [7]: The E-VQA dataset is derived from the VQAv2 [13] dataset, designed to evaluate the knowledge editing capabilities of multimodal large language models (MLLMs) within the context of visual question answering tasks. The dataset comprises 6,346 training samples and 2,093 testing samples, with evaluation metrics covering reliability, generality (T-Generality and M-Generality) and locality (T-Locality and M-Locality).

**E-IC** [7]: The E-IC dataset is constructed from the COCO Caption [5] dataset and serves as a benchmark for evaluating the knowledge editing capabilities of MLLMs in image caption tasks. The dataset includes 2,849 training samples and 1,000 testing samples. The evaluation framework mirrors that of E-VQA, thereby offering a robust means to gauge the effectiveness of knowledge editing in image captioning scenarios.

Each sample in E-VQA and E-IC datasets includes an edit instance, along with additional samples to evaluate textual and multimodal generality and locality. Generality samples are generated by rephrasing images and queries using tools like Stable Diffusion [34] and ChatGLM [10], ensuring the model's ability to generalize edits across different modalities. And locality samples are derived from unrelated images and queries from datasets such as OK-VQA [27], providing a stringent test of edit precision and locality. These datasets provide a comprehensive benchmark for evaluating model editors, focusing on their ability to maintain overall task performance while updating MLLM-specific factual knowledge.

**VLKEB** [17]: The Vision-Language Knowledge Editing Benchmark (VLKEB) is a large-scale dataset designed to evaluate the knowledge editing capabilities of MLLMs in realistic scenarios. Unlike prior benchmarks that primarily rely on synthetic or simplified data, VLKEB uses real-world images linked to structured knowledge triples from a multimodal knowledge graph. It contains 8,174 editing instances (5,000 for training and 3,174 for evaluation) and over 18,000 images, which provides a rigorous and comprehensive benchmark for advancing research in factual editing for MLLMs.

### A.2  Description of Baselines

**FT-L** [7] fine-tunes the last layer of the language model within MLLMs, which is the most widely used strategy for adapting pre-trained models to specific tasks.

**TP** [18] is an efficient model editing technique that implements knowledge updating by injecting small, trainable neurons (referred to as patches) into the final feed-forward layer. Each patch is tailored to correct a distinct mistake, allowing for precise, sequential edits without compromising the model's overall functionality.

**MEND** [30] trains a hypernetwork via meta-learning that enables efficient model editing by transforming back-propagated gradients into FFN matrix parameter offsets. It enables precise edits with a single example, providing a fast and scalable method for editing large language models.

**SERAC** [31] is a memory-based model editing method that stores edits externally and uses a scope classifier to selectively apply them via a counterfactual model. This allows for accurate, non-destructive updates without retraining, preserving the base model's original behavior.

**LEMoE** [39] is a model editing framework that enables continual, non-destructive updates to large language models by dynamically adding expert modules. It ensures stable routing with KV anchor mechanisms and optimizes edit order through clustering, effectively addressing forgetting and interference in lifelong editing scenarios.

**RECIPE** [3] is a lifelong model editing approach that encodes edited knowledge as continuous prompts and enhances large language models (LLMs) through a retrieval mechanism. Furthermore, it employs a knowledge sentinel to filter the edit samples for each query, enabling efficient and accurate updates to LLMs.

**LiveEdit** [2] is a specialized framework for lifelong knowledge editing in multimodal large language models (MLLMs). It employs a low-rank mixture-of-experts (MoE) approach, generating tailored low-rank experts for each specific edit. During inference, LiveEdit implements a two-stage routing mechanism: initially, it uses hard filtering based on visual semantics to discard irrelevant experts, followed by soft routing that leverages textual relevance to integrate the appropriate experts.

### A.3 Implementation Details

During training, we used the Adam optimizer with batch size of 8, learning rate of 1e-5, and set the number of epochs to 120. During testing, the threshold $\beta$ for the edit scope classifier was set to 0.4 for experiments on the E-IC dataset, and 0.7 for the E-VQA and VLKEB datasets. In explicit guidance, the relevance threshold $\eta$ was set to 6. To comprehensively evaluate the model's performance, we employed reliability, generality (including T-Generality and M-Generality), and locality (including T-Locality and M-Locality) accuracy as evaluation metrics. All experiments were conducted on a single NVIDIA H100 GPU.

Table 6: Average editing results across three datasets and two base MLLMs. $T$: Num Edits.

| Method | $T = 1$ | | | | | $T = 100$ | | | | | $T = 1000$ | | | | |
|---|---|---|---|---|---|---|---|---|---|---|---|---|---|---|---|
| | Rel. | T-Gen. | M-Gen. | T-Loc. | M-Loc. | Rel. | T-Gen. | M-Gen. | T-Loc. | M-Loc. | Rel. | T-Gen. | M-Gen. | T-Loc. | M-Loc. |
| FT-L | 70.2 | 67.8 | 64.6 | 92.6 | 90.2 | 60.2 | 56.9 | 56.1 | 73.6 | 61.8 | 51.8 | 48.1 | 49.8 | 52.0 | 52.6 |
| TP | 52.5 | 52.5 | 47.6 | 88.2 | 86.5 | 33.4 | 33.1 | 29.9 | 24.7 | 31.1 | 17.2 | 16.7 | 16.4 | 6.8 | 11.2 |
| MEND | 93.2 | 92.0 | 91.9 | 93.1 | 74.2 | 20.9 | 20.9 | 20.4 | 48.7 | 43.5 | 19.0 | 18.9 | 18.8 | 46.8 | 42.6 |
| SERAC | 88.8 | 84.5 | 85.4 | 78.5 | 22.5 | 68.5 | 62.5 | 64.5 | 58.6 | 15.5 | 63.2 | 57.3 | 60.4 | 56.3 | 15.4 |
| LEMoE | 93.9 | 92.3 | 89.9 | 94.8 | 80.8 | 51.9 | 48.3 | 47.5 | 74.7 | 51.2 | 38.5 | 35.2 | 35.4 | 70.6 | 51.3 |
| RECIPE | 88.7 | 85.3 | 86.0 | 96.9 | 91.1 | 68.5 | 63.6 | 66.7 | 93.4 | 88.2 | 61.7 | 57.1 | 59.9 | 91.2 | 86.6 |
| LiveEdit | 91.5 | 90.5 | 87.6 | 100.0 | **100.0** | 90.3 | 88.3 | 82.1 | 100.0 | **99.9** | 87.0 | 83.1 | 77.4 | 100.0 | **98.9** |
| CARML | **99.6** | **99.6** | **99.4** | **100.0** | 97.9 | **99.2** | **99.0** | **98.5** | **100.0** | 97.5 | **98.5** | **98.2** | **97.0** | **100.0** | 96.3 |

## B  Summary of Experimental Results

Table 6 summarizes the average performance across three datasets and two base MLLMs. It can be observed that CARML significantly outperforms other methods on most metrics, with only a slight underperformance in M-Locality compared to LiveEdit. Moreover, CARML maintains consistently high performance even as the number of edits (T) increases, demonstrating its robustness and superiority in lifelong multimodal editing scenarios.

## C  Pseudo Code of CARML

The pseudo-code of the CARML editing stage is in Algorithm 1, and the one of the CARML inference stage is Algorithm 2.

---

**Algorithm 1** CARML Editing Stage

**Input:** The initial multi-type edit knowledge base $K$, the edit dataset $\mathcal{D}_{\text{edit}}$ whose length is $T$.

**Output:** The final multi-type edit knowledge base $K = \{k_t = (e_v^t, e_x^t, e_m^t, o_t)\}_{t=1}^{T}$ after $T$ edits.

1: **for** each edit $(v_t, x_t, o_t) \in \mathcal{D}_{\text{edit}}, t \in [T]$ **do**
2:    *Get multi-type knowledge representations.*
3:    Convert $(v_t, x_t, o_t)$ to $e_v^t, e_x^t, e_m^t$ using Equation 3.3.
4:    *Update the multi-type edit knowledge base $K$.*
5:    $K_t = K_{t-1} \cup \{k_t = (e_v^t, e_x^t, e_m^t, o_t)\}$.
6: **end for**
7: **return** The final multi-type edit knowledge base $K = \{k_t = (e_v^t, e_x^t, e_m^t, o_t)\}_{t=1}^{T}$.

---

**Algorithm 2** CARML Inference Stage

---

**Input:** The MLLM model $f_\Theta$ including the embedding layer $f_{\text{emb}}$, the test dataset $\mathcal{D}_{\text{test}}$, the final multi-type edit knowledge base $K = \{k_t = (e_v^t, e_x^t, e_m^t, o_t)\}_{t=1}^T$, the trained edit scope classifier $f_{\text{cls}}$, the implicit guidance module $f_{\text{imp}}$, the edit scope threshold $\beta$, and the logits relevance threshold $\eta$.
**Output:** The model's output $o$.

1: **for** each edit sample $q = (v_q, x_q) \in \mathcal{D}_{\text{test}}$ **do**
2:     Convert $q = (v_q, x_q)$ to $q_e = (e_v^q, e_x^q, e_m^q)$ using Equation 3.3.
3:     Perform conflict-aware dynamic retrieval to pinpoint the most relevant knowledge $k_b = \left(e_v^b, e_x^b, e_m^b, o_b\right)$ in $K$.
4:     **if** $f_{\text{cls}}(q_e, k_b) > \beta$ **then**
5:         Generate static knowledge prompts $p_x$ and context-aware dynamic prompts $p_f$ with $f_{\text{imp}}$.
6:         Get the initial probability distribution $l = f_\Theta\left(p_x \oplus p_f \oplus f_{\text{emb}}\left(v_q\right) \oplus f_{\text{emb}}\left(x_q\right)\right)$.
7:         **if** $-\log\left(o_b \mid l\right) < \eta$ **then**
8:             Perform logits enhancement $l' = l + h(o_b)$.
9:             $o = \text{argmax}(l')$.
10:         **else**
11:             $o = \text{argmax}(l)$.
12:         **end if**
13:     **else**
14:         $o = f_\Theta\left(f_{\text{emb}}\left(v_q\right) \oplus f_{\text{emb}}\left(x_q\right)\right)$.
15:     **end if**
16: **end for**
17: **return** The model's output $o$.

---

# D  Description of Editing Loss

The editing loss is designed to ensure three key properties of knowledge editing: reliability, generality, and locality. Given a MLLM $f_\Theta$ and a training batch consisting of $b$ editing samples $(v_t, x_t, o_t)$, along with corresponding generality samples $(v_g, x_g, o_t)$ and locality samples $(v_l, x_l, o_l)$, the loss terms associated with these properties are defined as follows:

**Reliability Loss $\mathcal{L}_{rel}$:** This term serves to minimize the negative log-likelihood of generating the desired output $o_t$ on the editing samples $(v_t, x_t, o_t)$, conditioned on the static knowledge prompt $p_x$ and context-aware dynamic prompt $p_f$ generated in Section 3.4.1.

$$\mathcal{L}_{rel} = -\log f_\Theta\left(o_t \mid p_x \oplus p_f \oplus f_{emb}\left(v_t\right) \oplus f_{emb}\left(x_t\right)\right) \tag{11}$$

**Generality Loss $\mathcal{L}_{gen}$:** Beyond correcting individual specific inputs, the edited MLLM should generalize to semantically equivalent neighborhoods $(v_g, x_g, o_t)$. This loss is defined analogously to the reliability loss.

$$\mathcal{L}_{gen} = -\log f_\Theta\left(o_t \mid p_x \oplus p_f \oplus f_{emb}\left(v_g\right) \oplus f_{emb}\left(x_g\right)\right) \tag{12}$$

**Locality Loss $\mathcal{L}_{loc}$:** To ensure the edit remains localized, the output distribution of the edited model should remain close to that of the original frozen model $f_\Theta$ on unrelated samples $(v_l, x_l, o_l)$. This is enforced by minimizing the KL divergence between their outputs.

$$\mathcal{L}_{loc} = \text{KL}\left(f_\Theta\left(f_{emb}\left(v_l\right) \oplus f_{emb}\left(x_l\right)\right) \| f_\Theta\left(p_x \oplus p_f \oplus f_{emb}\left(v_l\right) \oplus f_{emb}\left(x_l\right)\right)\right) \tag{13}$$

The overall editing loss $\mathcal{L}_{edit}$ is the sum of these three components.

$$\mathcal{L}_{edit} = \mathcal{L}_{rel} + \mathcal{L}_{gen} + \mathcal{L}_{loc} \tag{14}$$

# E  Visualization Examples

In Figure 4, we present some additional visualization examples sourced from the E-IC, E-VQA, and VLKEB datasets. These diverse examples comprehensively demonstrate CARML's robust performance in achieving highly reliable, generalized, and localized multimodal knowledge editing.

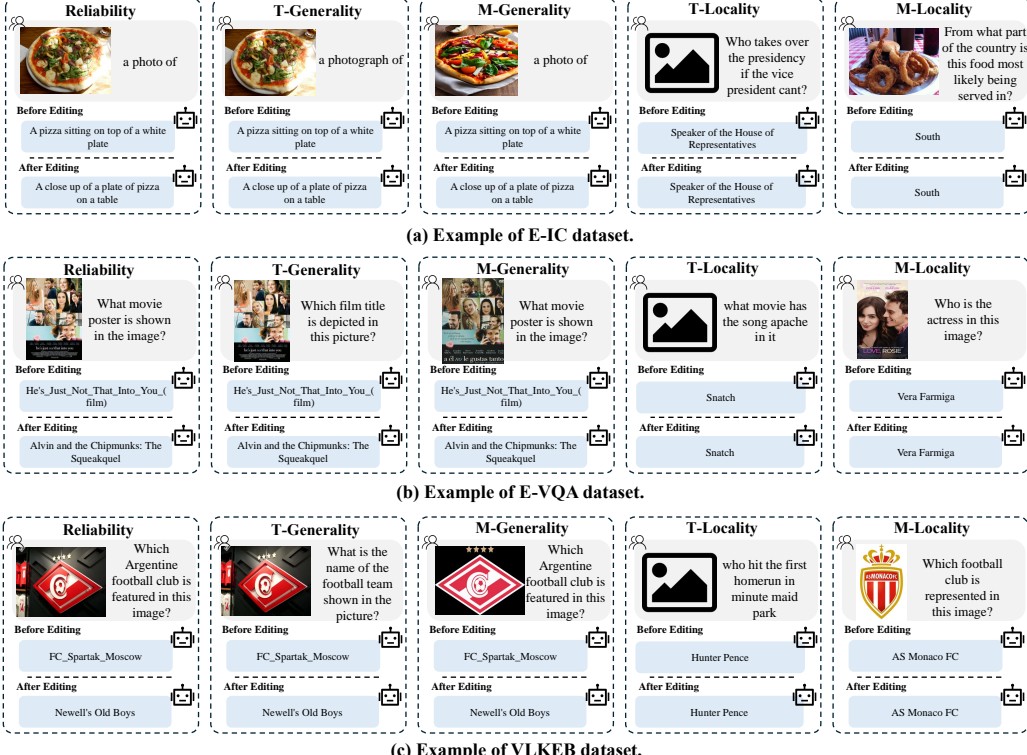

Figure 4: Visualization examples for multimodal knowledge editing.

# F  Broader Impacts

**Positive Societal Impacts.** Knowledge editing for MLLMs allows models to be updated quickly and precisely with new or corrected information, minimizing the need for full retraining. This capability supports timely responses in dynamic domains such as news, healthcare, and disaster response. By enabling fine-grained control over factual updates, these techniques can help curb the spread of outdated or incorrect information while improving model alignment with human values and current knowledge. Such advancements enhance trust and reliability in applications like education, accessibility tools, and interactive AI assistants.

**Negative Societal Impacts.** Despite its benefits, knowledge editing poses certain risks when misused or applied recklessly. Inaccurate or biased edits can propagate falsehoods with the same confidence and fluency as factual information, potentially leading to hallucinations or the spread of misinformation. Malicious actors could exploit these methods to implant targeted disinformation without leaving detectable traces in model parameters. Furthermore, without robust safeguards, knowledge editing could undermine model transparency and introduce new avenues for manipulation.

