# OpenReview forum: "Reliable Lifelong Multimodal Editing: Conflict-Aware Retrieval Meets Multi-Level Guidance"
_NeurIPS.cc/2025/Conference — NeurIPS 2025 poster_

### Official Review · Reviewer_eEUz · 2025-06-30

**Clarity:** 3
**Significance:** 2
**Originality:** 3
**Rating:** 3
**Confidence:** 3

**Summary:**

The paper presents CARML, a retrieval-augmented framework for lifelong multimodal knowledge editing in MLLMs. It tackles two key challenges: (1) precise retrieval from an ever-growing edit memory by quantifying both intra-modal uncertainty and inter-modal conflicts via a multi-channel softmax+Jaccard-based fusion; (2) reliable editing through a multi-level guidance strategy—implicit continuous prompts (static + context-aware dynamic) and an explicit logits-enhancement (“hard correction”) mechanism. An edit scope classifier further ensures locality by filtering out unrelated edits. Experiments on three benchmarks (E-VQA, E-IC, VLKEB) and two MLLMs demonstrate that CARML maintains near-100% reliability and generality even after 1,000 sequential edits, significantly outperforming baselines

**Questions:**

See the weakness

**Ethical Concerns:**

["NO or VERY MINOR ethics concerns only"]

**Final Justification:**

I reserve my opinion, but it is also OK if the paper is accepted.

**Limitations:**

yes

**Quality:**

2

**Strengths And Weaknesses:**

Strengths:

1. The combination of multi-channel retrieval, uncertainty/conflict modeling, and multi-level guidance (prompts + logits) is comprehensive and thoughtful.

2. The paper identifies and tackles scalability, accuracy, and edit locality issues that are relevant in real-world settings.

3. Results are reported across multiple datasets (E-VQA, E-IC, VLKEB) and strong baselines, showing consistent improvements.

Weaknesses:

1. The motivation is still confused. (1) The precise retrieval from an ever-growing edit memory. I do not know the concrete challenge in existing methods. Compared with existing methods, why can the proposed retrieval strategy solve this problem? In addition, the system seems complex. How do the benefits compare to the complexity? (2) As for the editing component, what is its advantage for the ever-growing knowledge?

2. While many modules are proposed (such as edit scope classifier, two-level prompts, logits correction), it is unclear which are critical for performance and which are incremental.

---

> ### Author Rebuttal · Authors · 2025-07-31
>
> We sincerely thank you for the valuable comments! Your comments are crucial for us to improve our work. We will address your concerns point by point.
>
> >**Q1:** The retrieval challenges faced by existing methods.
>
> **A1:** The retrieval strategies of existing methods face the following challenges:
>
> **(1) Ambiguity of text semantics:** Most current methods are designed for text-only knowledge editing and rely solely on text semantics for retrieval. However, due to the inherent ambiguity of text semantics (e.g., "a photo of" vs. "a photograph of" in the E-IC task), the lack of auxiliary visual information can easily lead to retrieval errors.
>
> **(2) Limitations of multimodal fusion strategies:** The first lifelong multimodal editing method, LiveEdit [1], retrieves multiple relevant experts through visual search and then fuses them using text semantic similarity weighting. However, this approach can introduce noise from irrelevant experts and cannot effectively handle direct conflicts between modalities.
>
> [1] Chen et al. Lifelong Knowledge Editing for Vision Language Models with Low-Rank Mixture-of-Experts, CVPR 2025.
>
> >**Q2:** Effectiveness of the proposed retrieval strategy.
>
> **A2:** CARML's conflict-aware dynamic retrieval strategy focuses on the unique contradictions present in multimodal editing. For instance, in the Generality-type editing samples, single-modality modifications often increase the risk of conflict. By retrieving all modalities in parallel and dynamically quantifying intra-modality uncertainty and inter-modality conflict, CARML can accurately identify and suppress unreliable modality channels, significantly improving retrieval accuracy. In Figure 1(a) of our paper, we illustrate the retrieval accuracy trends of various methods under an increasing number of sequential edits. The results show that CARML consistently maintains a desirable retrieval accuracy, proving its robustness.
>
> >**Q3:** The advantages of the proposed method compared to its complexity.
>
> **A3:** Thank you for raising this critical question. We believe that while CARML introduces a moderate level of complexity in its design, it delivers significant improvements in performance and reliability through three core advantages:
>
> **(1) Precise knowledge retrieval:** As stated in our response to Q2, our retrieval mechanism overcomes many shortcomings of existing methods, ensuring the accuracy of knowledge localization.
>
> **(2) Reliable multimodal knowledge editing:** Through a multi-level, coordinated editing guidance strategy, we ensure that the retrieved knowledge is correctly understood and executed by the model.
>
> **(3) Excellent performance-efficiency trade-off:** Although CARML introduces multiple modules to optimize performance, they are all of a lightweight design, primarily involving vector operations and small MLPs. We have discussed the comparative analysis of runtime efficiency in Section 4.3 of our paper. Specifically, we compare the accuracy and single editing time of the different methods after 1000 edits on the E-IC dataset, as shown in Table 1. The results demonstrate that CARML achieves an ideal efficiency ratio, attaining extremely high accuracy while maintaining a low time cost.
>
> Table 1: The trade-off between accuracy and editing time.
> | Method | FT-L | TP | MEND | SERAC | LEMoE | RECIPE | LiveEdit | CARML |
> |:---|:---:|:---:|:---:|:---:|:---:|:---:|:---:|:---:|
> | Editing time (s) | 1.5 | 5.27 | 0.125 | 0.117 | 1.344 | 0.073 | 0.113 | 0.141 |
> | Accuracy | 47.68 | 18.62 | 10.68 | 37.3 | 57.06 | 63.18 | 82.2 | 99.64 |
>
> >**Q4:** The advantages of the editing component for the ever-growing knowledge.
>
> **A4:** Our method has the following significant advantages when dealing with the ever-growing knowledge:
>
> **(1) Immunity to catastrophic forgetting:** CARML does not directly modify the core parameters of the MLLM. Instead, it guides the model through external prompts and logits enhancement. This fundamentally prevents the catastrophic forgetting problem that can arise from accumulating knowledge.
>
> **(2) Constant editing efficiency:** The computational overhead of CARML's editing component is independent of the knowledge base size. For each edit, the conflict-aware dynamic retrieval module precisely locates the Top-1 relevant knowledge item and passes this single item to the subsequent editing modules. This ensures that the efficiency of the editing component remains constant even as the knowledge base grows.
>
> >**Q5:** The division between critical modules and incremental modules.
>
> **A5:** Thank you for your valuable suggestion. Based on the ablation study in Table 4 of our paper and the additional ablation experiments shown below, we have re-evaluated the importance of the modules and categorized them as follows:
>
> Table 2: Ablation study on core components on the E-IC dataset.
> | Method | Rel. | T-Gen. | M-Gen. | T-Loc. | M-Loc. |
> | :--- | :---: | :---: | :---: | :---: | :---: |
> | w/o conflict-aware dynamic retrieval | 37.0 | 38.6 | 37.2 | 99.4 | 98.6 |
> | w/o edit scope classfier | 99.4 | 99.5 | 99.3 | 0.0 | 0.0 |
> | w/o context-aware prompt+logits | 70.2 | 73.4 | 69.4 | 100.0 | 100.0 |
> | w/o static knowledge prompt+logits | 80.5 | 77.4 | 80.9 | 100.0 | 100.0 |
> | w/o logits | 83.4 | 81.0 | 82.7 | 100.0 | 100.0 |
> | CARML | 99.4 | 99.5 | 99.3 | 100.0 | 100.0 |
>
> **(1) Critical Modules:**
>
> ● Conflict-aware dynamic retrieval: Significantly improves retrieval accuracy, leading to substantial enhancements in reliability and generality.
>
> ● Edit scope classifier: Effectively prevents modifications to irrelevant knowledge by learning reliable semantic boundaries.
>
> ● Context-aware dynamic prompt: Facilitates the effective execution of edits by providing highly customized, multimodal editing instructions.
>
> ● Output logits enhancement: Further boosts editing reliability, increasing it from 83.4% to 99.4%.
>
> **(2) Incremental Modules:**
>
> ● Static knowledge prompt: Although its removal results in a slight decline in overall performance, its essential information has been efficiently integrated into the "context-aware dynamic prompt" module.
>
> Thank you once again for your time and valuable feedback!

---

> > ### Comment · Reviewer_eEUz · 2025-08-03
> >
> > Thanks for your rebuttal. The RAG strategy is important, but I still do not understand its novelty and effectiveness. How does it solve the two challenges?

---

> > > ### Author Response · Authors · 2025-08-03
> > >
> > > Thank you for your further comments and for providing us with the opportunity to elaborate on the innovation and effectiveness of our retrieval strategy. We apologize that our previous explanation did not sufficiently convey the core concepts and will now offer a clearer response to your points of concern:
> > >
> > > **(1) Core Innovations:**
> > >
> > > Our retrieval strategy is centered on a multi-channel retrieval mechanism (visual, textual, and multimodal) and introduces a novel retrieval confidence assessment and dynamic fusion method. The specifics are as follows:
> > >
> > > ● Introduction of intra-modal uncertainty and inter-modal conflict: We quantify the uncertainty of a single retrieval channel using information entropy and measure the degree of conflict between different channels using Jaccard similarity. This allows for an accurate assessment of the confidence level of each retrieval channel.
> > >
> > > ● Dynamic fusion strategy: Based on the results of the confidence assessment, our method dynamically adjusts the weight of each channel, suppressing interference from low-confidence channels while amplifying the contribution of high-confidence ones.
> > >
> > > ● Precise localization and interference elimination: Ultimately, only the Top-1 retrieval result with the highest confidence is selected. This enables precise localization of the single most relevant piece of knowledge while effectively eliminating interference from irrelevant information.
> > >
> > > **(2) Addressing the Two Retrieval Challenges:**
> > >
> > > In response to the two key challenges, we have adopted the following strategies:
> > >
> > >
> > > ● Ambiguity of text semantics: To overcome the limitations of relying solely on text-based semantic retrieval, we introduced a multi-channel retrieval strategy. In addition to the textual semantic channel, we incorporated visual and multimodal channels to search the editing memory more comprehensively.
> > >
> > > ● Limitations of multimodal fusion strategies: Following the multi-channel retrieval, we innovatively introduce intra-modal uncertainty (assessed via information entropy) and inter-modal conflict (calculated via Jaccard similarity) to quantify the confidence of each channel. Based on this confidence assessment, we suppress interference from low-confidence channels, increase the weight of high-confidence channels, and dynamically fuse the multi-channel retrieval results. This strategy ensures that only the Top-1 most relevant piece of knowledge is retained, further reducing noise from irrelevant information.
> > >
> > > **(3) Validation of the RAG Retrieval Strategy's Effectiveness:**
> > >
> > > ● Performance Improvement: As shown by the ablation study in Table 4 of our paper, our conflict-aware dynamic retrieval strategy significantly improves the editing reliability and generality compared to the baseline text-only retrieval strategy, with the average of the three metrics improving from 37.6% to 71.3%, clearly demonstrating its performance advantage.
> > >
> > > ● Robustness: In Figure 1(a) of the paper, we illustrate the retrieval accuracy trends of different methods as the number of sequential edits increases. The results show that while the performance of other methods degrades with more edits, our CARML maintains a consistently high level of retrieval accuracy. This robustness is particularly crucial for large-scale, real-world applications.
> > >
> > > We hope this more direct and clear explanation will help you better understand the core innovation of our retrieval strategy and its effectiveness. If you have any further questions, please feel free to continue the discussion.
> > >
> > > Thank you again for your constructive review.

---

> > > ### Author Response · Authors · 2025-08-06
> > >
> > > We sincerely thank you for your time and valuable feedback. We hope our responses have addressed your concerns and helped clarify our contributions. If our clarifications resolve your doubts, we would be grateful for your reconsideration of the evaluation. Regardless, we truly appreciate your careful review and engagement with our work.

---

### Official Review · Reviewer_4PKH · 2025-07-02

**Clarity:** 2
**Significance:** 2
**Originality:** 2
**Rating:** 4
**Confidence:** 3

**Summary:**

In this paper, the authors propose CARML, a novel retrieval-augmented editing framework that integrates conflict-aware dynamic retrieval with multi-level implicit and explicit guidance for reliable lifelong multimodal editing. The authors introduce a conflict-aware dynamic retrieval mechanism that uniquely exploits multi-channel retrieval information for accurate knowledge localization based on quantified uncertainty and conflict degree. Besides, the authors construct an edit scope classifier, which guarantees the locality of editing by effectively filtering irrelevant edit samples. Next, the authors design an innovative multi-level collaborative guidance strategy, integrating implicit (static and dynamic prompts) and explicit (logits enhancement) knowledge injection to achieve reliable editing.

**Questions:**

NA

**Ethical Concerns:**

["NO or VERY MINOR ethics concerns only"]

**Final Justification:**

I think the authors have addressed my concern, and I think this paper could be accepted.

**Limitations:**

Yes

**Quality:**

2

**Strengths And Weaknesses:**

Strenght
1. The authors introduce CARML, a novel retrieval-augmented editing framework specifically designed to tackle the above challenges. The core innovation of CARML lies in its integration of conflict-aware dynamic retrieval and multi-level implicit and explicit knowledge guidance, enabling reliable lifelong knowledge editing in MLLMs.
2. The proposed method can achieve good results on MMedit benchmark

Weakness
1. Some recent proposed baselines are not compared, like AlphaEdit
2.  The underlying theory or mechanism are not mentioned， making the proposed method heuristic

---

> ### Author Rebuttal · Authors · 2025-07-31
>
> We sincerely thank you for the valuable comments! Your comments are crucial for us to improve our work. We will address your concerns point by point.
>
> >**Q1:** Lack of comparison with some recent baselines.
>
> **A1:** Thank you for your suggestion. We have further compared CARML with two of the latest baselines proposed in 2025: AlphaEdit [1] and BalancEdit [2]. We conducted experiments with sequential 1000 edits based on the BLIP-2 OPT model on the three datasets. The results are shown in the Table 1, 2 and 3. CARML's performance surpasses both of these recent baselines, further demonstrating its effectiveness. And we will supplement these baselines in the revision.
>
> Table 1: Editing results after 1000 edits on the E-IC dataset.
> | Method | Rel. | T-Gen. | M-Gen. | T-Loc. | M-Loc. |
> | :--- | :---: | :---: | :---: | :---: | :---: |
> | AlphaEdit | 60.5 | 54.9 | 50.7 | 57.2 | 50.3 |
> | BalancEdit | 70.5 | 68.4 | 57.2 | 58.1 | 51.5 |
> | CARML | 99.4 | 99.5 | 99.3 | 100.0 | 100.0 |
>
>
> Table 2: Editing results after 1000 edits on the E-VQA dataset.
> | Method | Rel. | T-Gen. | M-Gen. | T-Loc. | M-Loc. |
> | :--- | :---: | :---: | :---: | :---: | :---: |
> | AlphaEdit | 86.7 | 82.2 | 79.6 | 89.0 | 75.7 |
> | BalancEdit | 82.7 | 76.4 | 75.3 | 65.0 | 58.1 |
> | CARML | 97.0 | 95.9 | 93.2 | 100.0 | 91.9 |
>
>
> Table 3: Editing results after 1000 edits on the VLKEB dataset.
> | Method | Rel. | T-Gen. | M-Gen. | T-Loc. | M-Loc. |
> | :--- | :---: | :---: | :---: | :---: | :---: |
> | AlphaEdit | 89.4 | 82.7 | 84.1 | 86.4 | 78.3 |
> | BalancEdit | 79.6 | 76.2 | 70.9 | 62.5 | 63.7 |
> | CARML | 99.1 | 99.1 | 98.8 | 100.0 | 95.5 |
>
> [1] Fang et al. AlphaEdit: Null-Space Constrained Knowledge Editing for Language Models, ICLR 2025.
>
> [2] Guo et al. BalancEdit: Dynamically Balancing the Generality-Locality Trade-off in Multi-modal Model Editing, ICML 2025.
>
> >**Q2:** The underlying theory or mechanism are not mentioned, making the proposed method heuristic.
>
> **A2:** We thank the reviewer for highlighting the importance of the theoretical depth behind our method. The design of CARML is grounded in a decision-making framework that integrates multiple theories, with a controlled retrieval-augmented generation paradigm tailored for knowledge editing tasks at its core. Through non-invasive editing, CARML enhances the efficiency of multimodal knowledge editing while ensuring the stability required for lifelong learning. Specifically, its theoretical foundations include the following aspects:
>
> **(1) Information theory and ensemble learning:** To address the unique challenges of multimodal editing tasks, such as increased retrieval uncertainty and inter-modal conflicts arising from unimodal modifications in generality-type editing samples, we introduce information entropy to quantify retrieval uncertainty and use the Jaccard similarity to measure modal conflicts. Concurrently, drawing inspiration from ensemble learning, we treat the multimodal channels as "experts" and employ an adaptive mechanism to evaluate and fuse the retrieval results from different modalities.
>
> **(2) Metric learning:** To ensure the locality of edits, we have designed an edit scope classifier to learn a robust semantic decision boundary. This classifier effectively aligns the semantic spaces of the edit samples and the retrieved knowledge while filtering out samples that do not require editing, thereby guaranteeing the precision of the edits.
>
> **(3) Bayesian inference and prompt Learning:** Our proposed explicit-implicit guidance strategy is a concrete implementation of a Bayesian framework. Within this strategy, the implicit prompt guides the generation process by introducing strong prior knowledge, while the explicit logits enhancement further optimizes the probability distribution. This synergy calibrates the generation trajectory and improves the reliability of the edits.
>
> Grounded in these robust theoretical foundations, each component of CARML demonstrates a significant performance improvement in multimodal knowledge editing tasks, as detailed in the ablation studies in Table 4 of our paper. Furthermore, CARML possesses broad transferability potential for applications in other domains, such as fact-checking, hallucination mitigation, and efficient personalization. We will further emphasize these theoretical foundations and their significance in the revision.
>
> Thank you once again for your time and valuable feedback!

---

> ### Author Response · Authors · 2025-08-08
>
> Dear Reviewer 4PKH,
>
> Thank you so much for your insightful and constructive review of our manuscript. We are writing to respectfully follow up on our rebuttal and to see if you have any further questions.
>
> Following your invaluable suggestions, we have made substantial efforts to improve our paper. The key clarifications and analyses from our response are summarized as follows:
>
> ● Comparison with recent baselines: To address your concern about missing comparisons, we have conducted new experiments comparing our CARML against two very recent SOTA methods, AlphaEdit and BalancEdit. As shown in the new tables, our method demonstrates a significant performance advantage, further validating its effectiveness.
>
> ● Elaboration on theoretical foundations: In response to your point about the method appearing heuristic, we have detailed the underlying theoretical and mechanistic principles of our approach. We explained how CARML is grounded in information theory, ensemble learning, metric learning, and Bayesian inference, providing a solid theoretical basis for its design and success.
>
> We deeply value your expertise and hope that our responses and the new results have fully addressed your concerns. We would be very grateful to know if you have any remaining questions or if there's anything that requires further clarification. If our rebuttal has successfully resolved your initial concerns, we would be deeply appreciative if you would consider re-evaluating our paper.
>
> Many thanks for your constructive comments, time, and patience.
>
> Best regards,
>
> The Authors

---

### Official Review · Reviewer_9miH · 2025-07-02

**Clarity:** 3
**Significance:** 3
**Originality:** 3
**Rating:** 5
**Confidence:** 2

**Summary:**

This paper focuses on the lifelong multimodal editing problem. To address the problem, the paper proposed a novel retrieval-augmented editing framework designed for reliable lifelong multimodal knowledge editing, which consists of two parts: (1) Conflict-Aware Dynamic Retrieval and (2) Static and Dynamic Prompts. The Conflict-Aware Dynamic Retrieval component aims to locate the most relevant knowledge for the edit sample and determine whether to initiate the subsequent editing process, while the Static and Dynamic Prompts aim to provide both implicit guidance and explicit guidance to the editing process. The experiments across various datasets demonstrate the effectiveness of the proposed method.

**Questions:**

See above.

**Ethical Concerns:**

["NO or VERY MINOR ethics concerns only"]

**Final Justification:**

My concerns have been well addressed. Therefore, I keep my score.

**Limitations:**

Yes.

**Quality:**

3

**Strengths And Weaknesses:**

Strengths:
- The studied problem, lifelong multimodal model editing, is important to the community.
- The proposed method makes sense to me in general, although I am not in this field.
- The writing and the structure of the paper are good.
- The experiments are very comprehensive.

Weakness:
- Have you compared the runtime efficiency of your pipeline to other baselines? Since you mentioned that 'The dynamic nature of real-world information demands **efficient knowledge editing** in multimodal large language models, I think these experiments will also be good for your paper.
- The choice of models: BLIP 2-OPT and LLaVA-V1.5 (7B) are kind of outdated. I suggest the authors consider more recent models such as LLaVA-V1.6 and qwen 2.5 vl 7b instruct.

Overall: *I am not familiar with the domain. However, based on my reading, I think the paper is very interesting and complete, and the experiments are strong to support the claims made by the paper. However, there is some probability that I might miss some parts of the paper, or I might not fully understand the paper.*

---

> ### Author Rebuttal · Authors · 2025-07-31
>
> We sincerely thank you for the valuable comments! Your comments are crucial for us to improve our work. We will address your concerns point by point.
>
> >**Q1:** Comparison of runtime efficiency between CARML and other baselines.
>
> **A1:** Thank you for raising this important question. **Our proposed CARML demonstrates a superior trade-off between extremely high accuracy and low time cost.** We have discussed the comparative analysis of runtime efficiency in Section 4.3 of our paper, under the part of "Trade-off between Accuracy and Editing Time." Specifically, we compare the accuracy and single editing time of the different methods after 1000 edits on the E-IC dataset, and the results are shown in Table 1.
>
> Table 1: The trade-off between accuracy and editing time.
> | Method | FT-L | TP | MEND | SERAC | LEMoE | RECIPE | LiveEdit | CARML |
> |:---|:---:|:---:|:---:|:---:|:---:|:---:|:---:|:---:|
> | Editing time (s) | 1.5 | 5.27 | 0.125 | 0.117 | 1.344 | 0.073 | 0.113 | 0.141 |
> | Accuracy | 47.68 | 18.62 | 10.68 | 37.3 | 57.06 | 63.18 | 82.2 | 99.64 |
>
> >**Q2:** The MLLM models used are kind of outdated and suggest considering more recent models.
>
> **A2:** Thank you for your pertinent suggestion.
>
> **(1) Reason for choosing existing models:** We chose BLIP 2-OPT and LLaVA-V1.5 as the base models to ensure a fair comparison with existing knowledge editing methods, as these methods all use the same base models.
>
> **(2) Supplementary experiments:** To demonstrate the robustness of CARML, we conducted experiments on a more recent model, Qwen2.5-VL-7B-Instruct, as shown in Tables 2, 3 and 4. The results show that CARML also achieves excellent editing performance and surpasses existing baseline methods, proving CARML's model-agnostic nature and broad applicability.
>
> Table 2: Editing results after 1000 edits on the E-IC dataset.
> | Method | Rel. | T-Gen. | M-Gen. | T-Loc. | M-Loc. |
> | :--- | :---: | :---: | :---: | :---: | :---: |
> | FT-L | 51.4 | 42.7 | 40.6 | 59.6 | 57.0 |
> | LeMoE | 42.8 | 37.4 | 40.1 | 74.0 | 82.7 |
> | LiveEdit | 78.4 | 74.3 | 64.3 | 100.0 | 99.9 |
> | CARML | 99.4 | 99.3 | 99.4 | 100.0 | 100.0 |
>
>
> Table 3: Editing results after 1000 edits on the E-VQA dataset.
> | Method | Rel. | T-Gen. | M-Gen. | T-Loc. | M-Loc. |
> | :--- | :---: | :---: | :---: | :---: | :---: |
> | FT-L | 48.5 | 44.7 | 43.1 | 51.4 | 38.9 |
> | LeMoE | 38.8 | 26.0 | 30.4 | 68.7 | 38.4 |
> | LiveEdit | 91.7 | 89.3 | 81.6 | 100.0 | 96.7 |
> | CARML | 97.2 | 95.9 | 92.9 | 100.0 | 94.1 |
>
>
> Table 4: Editing results after 1000 edits on the VLKEB dataset.
> | Method | Rel. | T-Gen. | M-Gen. | T-Loc. | M-Loc. |
> | :--- | :---: | :---: | :---: | :---: | :---: |
> | FT-L | 62.7 | 54.9 | 58.3 | 68.0 | 60.2 |
> | LeMoE | 67.5 | 59.4 | 64.9 | 51.3 | 47.4 |
> | LiveEdit | 90.8 | 86.2 | 79.4 | 100.0 | 100.0 |
> | CARML | 99.3 | 99.1 | 98.8 | 100.0 | 95.5 |
>
>
> Thank you once again for your time and valuable feedback!

---

> > ### Comment · Reviewer_hMxv · 2025-08-05
> >
> > Thank you for the rebuttal! It helped clarify my questions. I will be keeping my score.

---

> ### Author Response · Authors · 2025-08-06
>
> Thank you for your rating. Your valuable suggestions greatly contribute to the quality of our manuscript. Thank you again for your precious time and valuable suggestions!

---

### Official Review · Reviewer_hMxv · 2025-07-03

**Clarity:** 2
**Significance:** 3
**Originality:** 3
**Rating:** 5
**Confidence:** 3

**Summary:**

This paper tackles the problem of lifelong knowledge editing for mLLMs. The proposed method, CARML, combines conflict-aware dynamic retrieval with multi-level knowledge guidance (static/dynamic prompts plus logits enhancement). The authors show that CARML maintains strong reliability and locality even as the number of edits grows, outperforming various baselines.

**Questions:**

1. Could you clarify if the conflict-aware retrieval is truly novel compared to general multi-channel retrieval and fusion techniques? What makes it fundamentally better?

2. What would the authors argue to be the most important component in the pipeline?

3. Given the pipeline complexity, how sensitive is the method to the scope classifier’s threshold or the retrieval fusion parameters? Any tips for tuning these?

4. Could you add a discussion about scaling the knowledge base — how will CARML handle millions of edits in practice?

**Ethical Concerns:**

["NO or VERY MINOR ethics concerns only"]

**Final Justification:**

The authors have addressed my concerns. In particular, they have added experiments for parameter sensitivity, additional ablation studies, etc.
My original biggest concern was that the paper had too many components in it and I remember taking some digging in order to make sense of the pipeline. In rebuttal, the authors had a clear idea of what they believed is the most important component, and I trust them to be. able to incorporate it into the final version.

**Limitations:**

Yes — the authors do a reasonable job discussing potential forgetting and the limits of current memory design. Might be good to add more about retrieval scalability in real deployments.

**Quality:**

3

**Strengths And Weaknesses:**

Strengths:

1. The paper is well-motivated -- lifelong editing for MLLMs is both timely and practically important! It also highlights the key chllanges of growing edit knowledge bases and preserving locality, which is well-aligned with the contributions.
2. The proposed solution is thorough and demonstrates impressive empirical results across multiple benchmarks and baselines, with ablation studies that justify each module.
3.  Strong performance even with large numbers of edits (e.g., 1000+) shows good scalability. I especially enjoy reading the experiments and analysis results, they've proactively answered most of the questions I have.

Weaknesses:
My biggest concern is that the proposed pipeline is quite complex, and the paper’s description can be hard to follow on a first read. Simplifying the narrative and emphasizing the key innovations more clearly would improve clarity.

In terms of better highlighting the key innovation within this pipeline:
1. Among the main steps, I feel the "Conflict-Aware Dynamic Retrieval" seems to be more or less standard (e.g., I think people in general would do alignment and matching between different modalities), but I do like the idea of Context-Aware Dynamic Prompt.

2. It remains a bit unclear which modules are absolutely necessary for the gains. While sec 4 shows a clear ablation studies gradually adding more components, an experiment isolating only the core parts (e.g., just context-aware dynamic prompt + logits bias) would make the impact of each clearer.

---

> ### Author Rebuttal · Authors · 2025-07-31
>
> We sincerely thank you for the valuable comments! Your comments are crucial for us to improve our work. We will address your concerns point by point.
>
> >**Q1:** Given the pipeline complexity, the clarity could be further improved.
>
> **A1:** Thank you for your pertinent suggestion.
>
> **(1) Complexity clarification:** Although CARML introduces multiple modules to optimize performance, they are all of a lightweight design, primarily involving vector operations and small MLPs, which results in extremely low computational overhead. In Section 4.3 of our paper, we illustrate the trade-off between performance and computational efficiency for CARML, showing that it achieves an excellent balance.
>
> **(2) Narrative improvement:** In the revision, we will highlight the key innovations of our method in the introduction. And in the methodology section, we will further refine the descriptive logic of each module to ensure its innovative aspects are articulated with clarity and coherence.
>
> >**Q2:** Conduct ablation studies on only core parts to identify the absolutely necessary modules.
>
> **A2:** Thank you for your valuable comments. Based on your recommendation, we have conducted an additional ablation study on the core components. The results are shown in the table below:
>
> Table 1: Ablation study on core components on the E-IC dataset.
> | Method | Rel. | T-Gen. | M-Gen. | T-Loc. | M-Loc. |
> | :--- | :---: | :---: | :---: | :---: | :---: |
> | w/o conflict-aware dynamic retrieval | 37.0 | 38.6 | 37.2 | 99.4 | 98.6 |
> | w/o edit scope classfier | 99.4 | 99.5 | 99.6 | 0.0 | 0.0 |
> | w/o context-aware prompt+logits | 70.2 | 73.4 | 69.4 | 100.0 | 100.0 |
> | w/o static knowledge prompt+logits | 80.5 | 77.4 | 80.9 | 100.0 | 100.0 |
> | w/o logits | 83.4 | 81.0 | 82.7 | 100.0 | 100.0 |
> | CARML | 99.4 | 99.5 | 99.3 | 100.0 | 100.0 |
>
> We can observe that the key modules that bring major performance improvements to CARML are: conflict-aware dynamic retrieval, editing scope classifier, context-aware dynamic prompt, and logits enhancement. In comparison, the contribution of the static knowledge prompt is relatively minor, as the information it carries is already encompassed by the context-aware dynamic prompt.
>
> >**Q3:** Regarding the innovation and advantages of conflict-aware dynamic retrieval.
>
> **A3:** The innovation and advantages of the "conflict-aware dynamic retrieval" mechanism are reflected in the following aspects:
>
> **(1) Dynamic adaptivity:** This mechanism dynamically allocates modality weights based on intra-modality uncertainty and inter-modality conflict, enabling precise adjustments for each editing sample.
>
> **(2) Customized for multimodal editing task:** It addresses the unique challenges of multimodal editing (e.g., the increased conflict caused by single-modality modifications in Generality-type editing samples) by capturing conflicts and dynamically reducing the weights of the relevant modalities.
>
> **(3) Lightweight and efficient:** The mechanism only involves vector operations and does not require computationally expensive re-rank models. Furthermore, the ablation study in **Q2** further validates the value of this mechanism, where the edit reliability is reduced from 99.4% to 37.0% after its elimination.
>
> >**Q4:** What is the most important component in the pipeline?
>
> **A4:** We believe the most important component is the multi-level knowledge guidance module, which ensures the effective execution of the knowledge edit. The implicit context-aware dynamic prompt generates highly customized, fine-grained multimodal editing instructions for the model, while the explicit logits enhancement further ensures the reliability of the editing outcome.
>
> >**Q5:** Regarding hyperparameter sensitivity and tuning techniques.
>
> **A5:** We have analyzed the key hyperparameters:
>
> **(1) Scope classifier's threshold β:** This threshold directly determines whether an editing operation is triggered and is therefore relatively sensitive, as shown in Table 2. Since the classifier has learned a stable semantic boundary, its decision threshold typically stabilizes around 0.5. We use a simple binary search method to fine-tune the value around β, selecting the optimal parameter that maximizes reliability without harming locality.
>
> Table 2: Sensitivity of the scope classifier's threshold β on the E-IC dataset.
> | β | Rel. | T-Gen. | M-Gen. | T-Loc. | M-Loc. |
> | :--- | :---: | :---: | :---: | :---: | :---: |
> | 0.2 | 99.9 | 99.7 | 99.6 | 96.4 | 95.3 |
> | 0.3 | 99.9 | 99.6 | 99.6 | 98.2 | 97.1 |
> | 0.4 | 99.4 | 99.5 | 99.6 | 100.0 | 100.0 |
> | 0.5 | 87.8 | 92.7 | 86.1 | 100.0 | 100.0 |
> | 0.7 | 49.4 | 64.0 | 45.4 | 100.0 | 100.0 |
>
>
> **(2) Retrieval fusion hyperparameters:** CARML exhibits low sensitivity to these parameters. For the fusion weights of uncertainty and conflict in Equation (5), experiments show that values within the range of [0.3, 0.7] achieve desirable results. For the Jaccard similarity K value, the experimental results of its sensitivity analysis are shown in Table 3. We can notice that its performance is stable for values in the range of [12, 21]. The K value that is too small may result in insufficient candidate samples, while one that is too large can introduce noise.
>
> Table 3: The sensitivity analysis on the value of K in the Jaccard similarity.
> | K | 3 | 9 | 12 | 15 | 18 | 21 | 30 |
> | :---: | :---: | :---: | :---: | :---: | :---: | :---: | :---: |
> | Recall@1 | 93.2% | 97.5% | 99.2% | 99.4% | 99.1% | 98.6% | 94.8% |
>
> >**Q6:** The scalability of CARML's knowledge base in real-world deployment.
>
> **A6:** Thank you for raising this practical and important question. Our CARML method possesses the following advantages for scalability:
>
> **(1) Efficient retrieval scalability:** Our retrieval mechanism is built upon an efficient vector similarity search library (FAISS). Its search time complexity has a sub-linear relationship with the knowledge base size N. Therefore, CARML can be easily extended to millions of editing entries, with the primary bottleneck being storage rather than retrieval latency.
>
> **(2) Constant editing overhead:** Since the conflict-aware dynamic retrieval module can precisely locate Top-1 relevant knowledge item, the amount of knowledge received in the editing phase is fixed at one, and its computational complexity is independent of the size of the knowledge base and remains constant.
>
> **(3) Immunity to catastrophic forgetting:** Unlike methods that directly modify model parameters, CARML stores knowledge externally and selects relevant entries via its retrieval mechanism. This approach avoids the problem of catastrophic forgetting that can arise from frequent iterative edits, ensuring the long-term stability of the model's capabilities.
>
> Thank you once again for your time and valuable feedback!

---

> ### Author Response · Authors · 2025-08-08
>
> We sincerely thank you for your recognition of our work and your valuable time. Your insightful and constructive feedback is crucial in helping us improve our paper.

---

### Official Review · Reviewer_fHJZ · 2025-07-07

**Clarity:** 2
**Significance:** 3
**Originality:** 2
**Rating:** 5
**Confidence:** 4

**Summary:**

This paper introduces CARML, a novel retrieval-augmented framework for lifelong knowledge editing in Multimodal Large Language Models (MLLMs). The authors identify two primary challenges in existing methods: imprecise knowledge retrieval and a lack of coordinated, multi-level guidance for the editing process. CARML addresses these issues through two core innovations. The first is a conflict-aware dynamic retrieval mechanism that constructs a multi-modal knowledge base and uses a sophisticated fusion strategy—quantifying both intra-modal uncertainty and inter-modal conflict—to accurately retrieve the most relevant historical edit for a new query. The second is a multi-level guidance strategy that combines implicit guidance (via learnable static and context-aware dynamic prompts) and explicit guidance (via a direct logits enhancement mechanism) to steer the MLLM toward the correct output. The authors conduct extensive experiments on several MLLM editing benchmarks (E-VQA, E-IC, VLKEB) using BLIP-2 and LLaVA models, demonstrating that CARML significantly outperforms a wide range of state-of-the-art baselines, maintaining near-perfect performance even after a thousand sequential edits.

**Questions:**

Regarding the conflict-aware fusion in your retrieval module, could you provide more justification for the specific formula used in Equation 5? Did you experiment with other methods for combining the uncertainty and conflict scores, and if so, how did they perform? How sensitive is the performance to the value of K used for calculating the Jaccard similarity?

**Ethical Concerns:**

["NO or VERY MINOR ethics concerns only"]

**Final Justification:**

After reading the rebuttal, all my questions have been properly addressed. Therefore, I raise my score.

**Limitations:**

Yes

**Quality:**

3

**Strengths And Weaknesses:**

Pros:
1. The paper is well-written, methodologically sound, and addresses a significant and timely problem in the field of MLLMs. The motivation is clear, and the proposed solution is both technically sophisticated and elegant.
2. Another significant strength is the multi-level knowledge guidance system. The combination of implicit and explicit guidance is compelling. The context-aware dynamic prompt, which uses the edit sample itself as a query to attend to different aspects of the retrieved knowledge, is a particularly clever mechanism for creating fine-grained, tailored instructions for the model. This moves beyond simply prepending static information and allows for a more nuanced interaction between the query and the stored knowledge.

Cons:
1. A key contribution of this paper is the Conflict-Aware Fusion and Re-ranking mechanism. The use of information entropy to quantify the internal uncertainty of retrieval candidates is an interesting approach. However, the idea of using entropy as an uncertainty signal to re-rank retrieved information has been explored in prior knowledge editing work [1]. To better situate your contribution, it would be beneficial to cite this related work and explicitly differentiate your method. Your primary innovation appears to be the novel combination of this intra-modal uncertainty with a measure of inter-modal conflict, which is tailored to the multimodal domain. Clarifying this distinction would strengthen the paper's claim of novelty.
2. I also have concerns around the "Explicit Guidance: Output Logits Enhancement" mechanism. The ablation study (Table 4) reveals that this single component is responsible for the final, dramatic leap in performance, pushing reliability from 83.4% to 99.4%. While undeniably effective, this mechanism feels less like "guiding" the model and more like a "hard correction" or a post-processing step that forces the model to output the cached answer. This raises questions about the nature of the "editing" being performed. Does the model truly learn to integrate the new knowledge into its parametric understanding, or does it simply learn to pass through the retrieved answer when triggered? This could potentially harm the model's generative fluency and its ability to reason about the edited concept. The paper would be strengthened by a more critical discussion of this component and its implications.

[1] Shi, Y., Tan, Q., Wu, X., Zhong, S., Zhou, K., & Liu, N. (2024, October). Retrieval-enhanced knowledge editing in language models for multi-hop question answering. CIKM 2024.

---

> ### Author Rebuttal · Authors · 2025-07-31
>
> We sincerely thank you for the valuable comments! Your comments are crucial for us to improve our work. We will address your concerns point by point.
>
> >**Q1:** The difference in the use of uncertainty for "Conflict-Aware Fusion and Re-ranking" mechanism compared to work [1].
>
> **A1:** Thank you for pointing out the related work [1]. The main differences between our CARML and work [1] are as follows:
>
> **(1) Different task types:** The work [1] focuses on single-modality (text) knowledge editing, whereas our work is centered on the field of lifelong multimodal knowledge editing.
>
> **(2) Different usage stages and objectives**: The work [1] uses mutual information maximization to retrieve unique knowledge subgraph, which is later pruned by analyzing the uncertainty generated by the LLM. In contrast, our approach intervenes at the retrieval stage to identify the most relevant knowledge by evaluating the uncertainty of the retrieval results across modalities.
>
> **(3) Our core Innovation:** We uniquely introduce the critical dimension of "inter-modality conflict." By combining "intra-modality uncertainty" with "inter-modality conflict," CARML can dynamically adjust the weights of each modality, thereby achieving precise knowledge localization.
>
> We will cite and discuss work [1] in detail in the revision to more clearly highlight the novelty of our work.
>
> >**Q2:** Concerns about the "Explicit Guidance: Output Logits Enhancement" mechanism.
>
> **A2:** Thank you for this valuable comment. We would like to clarify that the explicit guidance builds upon the parametric understanding established by the implicit prompt guidance. The implicit prompt is responsible for deriving a high-quality prior distribution of the editing knowledge. **The explicit guidance then calibrates this distribution by incorporating the target answer, further boosting the generation confidence for tokens relevant to the edit.**
>
> To verify this, we conducted a supplementary experiment: we selected 100 E-VQA samples that were incorrectly predicted with only implicit guidance. During the auto-regressive generation process, we applied logits enhancement only to the first token of the answer sequence and observed whether it improves the model's prediction accuracy for subsequent tokens. The results are as follows:
>
> Table 1: The accuracy of subsequent tokens with logits enhancement applied only to the first answer token.
>
> | Method | Rel. | T-Gen. | M-Gen. | T-Loc. | M-Loc. |
> | :--- | :---: | :---: | :---: | :---: | :---: |
> | w/o Logits | 0.0 | 2.7 | 0.0 | 100.0 | 96.5 |
> | + Logits | 31.6 | 32.0 | 28.9 | 100.0 | 94.3 |
>
> The experiment shows that the "Output Logits Enhancement" can enhance the output probability for editing knowledge-related tokens and help calibrate the generation trajectory.
>
> >**Q3:** Regarding the specific justification for Equation (5) and attempts with other fusion methods.
>
> **A3:** Thank you for your attention to the details of our formula. The purpose of Equation (5) is to reward modalities that exhibit low uncertainty and low inter-modality conflict. We achieve this as follows:
>
> (1) We quantify the contribution of each modality's uncertainty and conflict degree by subtracting the normalized entropy and conflict score using 1, respectively.
>
> (2) We assign equal weights to both factors. Experiments have shown that this simple weighting strategy demonstrates strong stability and generalization capability across different datasets.
>
> (3) We did experiment with introducing learnable parameters (a small MLP) to combine the uncertainty and conflict scores to predict modality weights. However, the introduction of additional training parameters easily led to overfitting, whereas the simple averaging approach proved to be more robust in our experiments.
>
> >**Q4:** Regarding the sensitivity of the K value in Jaccard similarity.
>
> **A4:** We conducted a sensitivity analysis on the value of K in the Jaccard similarity, using retrieval accuracy (Recall@1) as the metric. The experimental results on the E-IC dataset are as follows:
>
> Table 2: The sensitivity analysis on the value of K in the Jaccard similarity.
> | K | 3 | 9 | 12 | 15 | 18 | 21 | 30 |
> | :---: | :---: | :---: | :---: | :---: | :---: | :---: | :---: |
> | Recall@1 | 93.2% | 97.5% | 99.2% | 99.4% | 99.1% | 98.6% | 94.8% |
>
> The results indicate that the retrieval accuracy shows good stability when the value of K is within the range of [12, 21]. A K value that is too small can affect the conflict judgment due to insufficient candidate samples, while a value that is too large can introduce noise, leading to a decline in performance. We will add a complete discussion on hyperparameter sensitivity analysis in the revision.
>
> [1] Shi et al. Retrieval-enhanced knowledge editing in language models for multi-hop question answering. CIKM 2024.
>
> Thank you once again for your time and valuable feedback!

---

> > ### Comment · Area_Chair_rrGy · 2025-08-05
> > **Discussion**
> >
> > Hi Reviewer,
> >
> > The authors have provided the rebuttal. What are your thoughts on the response? Please engage in the discussion with the authors as soon as possible, as the deadline for discussion is August 8th.
> >
> > Thanks,
> >
> > AC

---

> ### Author Response · Authors · 2025-08-08
>
> Dear Reviewer fHJZ,
>
> Thank you very much for your thoughtful and constructive feedback. We are writing to respectfully follow up on our rebuttal and to see if you have any further questions.
>
> Following your invaluable suggestions, we have made substantial efforts to improve our manuscript. The key clarifications and analyses from our response are summarized as follows:
>
> ● To better situate our contribution, we have detailed the distinction between our conflict-aware fusion and re-ranking mechanism and prior work [1].
>
> ● To address your valid concern about our "output logits enhancement" mechanism, we conducted a new supplementary experiment. The results demonstrate that this mechanism can further calibrate the model's generation trajectory rather than simply forcing the output.
>
> ● We have provided a detailed justification for our Equation 5. Additionally, following your suggestion, we conducted a new sensitivity analysis on the hyperparameter K to further enhance the manuscript's technical completeness.
>
> We deeply value your expertise and hope that our responses and the new results have fully addressed your concerns. We would be very grateful to know if you have any remaining questions or if there's anything that requires further clarification.
>
> Many thanks for your constructive comments, time, and patience.
>
> Best regards,
>
> The Authors

---

### Note · Authors · 2025-08-12

Dear AC and reviewers,

We thank the AC and all reviewers for their valuable time and constructive feedback on our paper. We are very encouraged that the reviewers generally recognized the following points:

● The addressed problem of lifelong multimodal editing is **an important and timely topic** in the field of MLLMs. *(Reviewers fHJZ, hMxv, 9miH)*

● The proposed CARML framework, which combines conflict-aware dynamic retrieval with multi-level knowledge guidance, is **a novel and comprehensive solution**. *(Reviewers fHJZ, hMxv, 9miH, 4PKH)*

● The **experimental evaluation is sufficient**, demonstrating the superiority, reliability, and robustness of CARML in large-scale sequential editing. *(All reviewers)*

● The paper is **well-written, with clear motivation and good structure**. *(Reviewers fHJZ, 9miH)*

In response to the main concerns raised by the reviewers, we have conducted extensive supplementary experiments and analyses to further validate the effectiveness of CARML:

**(1) Methodological Novelty, Advantages, and Theoretical Foundation:**

● We clarify the core innovation and advantages of the "conflict-aware dynamic retrieval" mechanism, which addresses the unique challenges of lifelong multimodal editing by integrating "intra-modal uncertainty" and "inter-modal conflict”, significantly improving retrieval accuracy.

● We supplement the theoretical foundations of CARML, which are based on information theory, ensemble learning, metric learning, and Bayesian inference.

**(2) Analysis of Key Modules:**

● New experiments verify that the logits enhancement mechanism further calibrates the generation trajectory rather than performing a simple hard correction.

● More detailed ablation studies are provided to clearly distinguish between critical and incremental modules.

**(3) Expansion of Experimental Comparisons:**

● We include new comparative experiments against the latest SOTA methods (AlphaEdit, BalancEdit) to further demonstrate CARML's superiority.

● Additional experiments on a more advanced model (Qwen2.5-VL-7B-Instruct) validate the broad applicability of our method.

● We add analysis of hyperparameter sensitivity, as well as discussions on runtime efficiency and the scalability for large-scale deployment.

These experiments will be integrated into the main body or the appendix of our paper. Many thanks to the AC and all reviewers for their time, patience, and valuable comments!

Sincerely,

The Authors

---

### Decision · Program_Chairs · 2025-09-17

**Decision:**

Accept (poster)

**Comment:**

This paper introduces CARML, a novel retrieval-augmented framework for lifelong multimodal knowledge editing in MLLMs. Its core contributions include a conflict-aware dynamic retrieval mechanism that quantifies intra-modal uncertainty and inter-modal conflict, and a multi-level guidance strategy combining implicit prompts and explicit logits enhancement. Strengths include strong empirical performance across multiple benchmarks (E-VQA, E-IC, VLKEB), scalability to 1000+ edits, and a well-motivated design addressing real-world challenges. Weaknesses initially included concerns about pipeline complexity and the novelty of retrieval mechanisms.

During the rebuttal period, reviewers raised concerns about novelty compared to prior work, component necessity, hyperparameter sensitivity, and theoretical grounding. Authors responded thoroughly: they differentiated their retrieval method from unimodal approaches, provided ablation studies isolating core modules (e.g., conflict-aware retrieval and logits enhancement are critical), added sensitivity analyses for hyperparameters like K and β, and expanded comparisons to recent SOTA methods (AlphaEdit, BalancEdit). They also grounded their method in information theory, metric learning, and Bayesian inference, addressing heuristic concerns.

This paper is recommended for acceptance due to its significant empirical contributions, methodological novelty in multimodal conflict-aware retrieval, and comprehensive rebuttal addressing all major concerns. The authors demonstrated scalability, robustness across models (including Qwen2.5-VL), and superior performance over strong baselines.